# Therapeutic benefits of maintaining CDK4/6 inhibitors and incorporating CDK2 inhibitors beyond progression in breast cancer

Jessica Armand[1,2], Sungsoo Kim[1,2], Kibum Kim[1], Eugene Son[1], Minah Kim[1,2], Kevin Kalinsky[3], Hee Won Yang[1,2]*

[1]Department of Pathology and Cell Biology, Columbia University Irving Medical Center, New York, United States; [2]Herbert Irving Comprehensive Cancer Center, Columbia University Irving Medical Center, New York, United States; [3]Winship Cancer Institute at Emory University, Department of Hematology and Medical Oncology, Atlanta, United States

*For correspondence:
hy2602@cumc.columbia.edu

Competing interest: The authors declare that no competing interests exist.

## eLife Assessment

This study presents **fundamental** insights into overcoming resistance in hormone receptor-positive breast cancer by demonstrating that sustained CDK4/6 inhibitor treatment, either alone or in combination with CDK2 inhibitors, significantly suppresses the growth of drug-resistant cancer cells. The findings are supported by **compelling** evidence from both in vitro cell line experiments and in vivo mouse models, highlighting the therapeutic potential of maintaining CDK4/6 inhibitors beyond disease progression. Additionally, the identification of cyclin E overexpression as a key driver of resistance offers a target that will be of value for future therapeutic strategies, potentially improving outcomes for patients with advanced breast cancer.

**Abstract** CDK4/6 inhibitors (CDK4/6i) with endocrine therapy are standard for hormone receptor-positive (HR⁺) metastatic breast cancer. However, most patients eventually develop resistance and discontinue treatment, and there is currently no consensus on effective second-line strategies. Using preclinical HR⁺ human breast cancer models with acquired resistance to CDK4/6i, we demonstrate that maintaining CDK4/6i therapy, either alone or combined with CDK2 inhibitors (CDK2i), slows the growth of resistant tumors by prolonging G1 progression. Mechanistically, sustained CDK4/6 blockade in drug-resistant cells reduces E2F transcription and delays G1/S via a noncanonical, posttranslational regulation of retinoblastoma protein (Rb). Durable suppression of both CDK2 activity and growth of drug-resistant cells requires co-administration of CDK2i with CDK4/6i. Moreover, cyclin E overexpression drives resistance to the combination of CDK4/6i and CDK2i. These findings elucidate how continued CDK4/6 blockade constrains resistant tumors and support clinical strategies that maintain CDK4/6i while selectively incorporating CDK2i to overcome resistance.

## Introduction

Metastatic breast cancer remains a leading cause of cancer-related mortality in women globally (*Lei et al., 2021*; *Houghton and Hankinson, 2021*). A key dysregulation in breast cancer involves the

overactivation of cyclin-dependent kinases 4 and 6 (CDK4/6) (*Sherr et al., 2016*; *Watt and Goel, 2022*; *Fassl et al., 2022*; *Shanabag et al., 2025*). Active CDK4/6 phosphorylates the retinoblastoma protein (Rb), a crucial regulator that prevents cell-cycle initiation by sequestering E2F transcription factors (*Engeland, 2022*). Rb phosphorylation results in the release of E2F, thereby promoting CDK2 activation and cell proliferation (*Kim et al., 2022*; *Fisher, 2016*). Understanding this mechanism has driven significant advancements in therapeutic strategies, particularly for hormone receptor-positive (HR⁺)/human epidermal growth factor receptor 2-negative (HER2⁻) breast cancer, which constitutes approximately 70% of breast cancer cases (*Howlader et al., 2014*). The current standard first-line treatment for HR⁺/HER2⁻ metastatic breast cancer is a combination of CDK4/6 inhibitors (CDK4/6i) and endocrine therapy (ET) (*Watt and Goel, 2022*; *Fassl et al., 2022*; *Goel et al., 2022*). Although this strategy has significantly improved patient outcomes, resistance remains a major challenge: approximately 30% of patients develop resistance within 2 years, and the majority ultimately relapse (*Hortobagyi et al., 2016*; *Johnston et al., 2019*). Upon disease progression, CDK4/6i therapy is typically discontinued, often resulting in aggressive tumor regrowth and a lack of effective second-line treatment options.

While continuing ET after disease progression confers clinical benefit (*Robertson et al., 2005*; *Vergote et al., 2003*; *Xie et al., 2019*; *Steger et al., 2005*; *Barrios et al., 2012*), the value of maintaining CDK4/6i treatment beyond progression remains elusive. Multiple ongoing and completed trials have evaluated continuing the FDA-approved CDK4/6 inhibitors, palbociclib, abemaciclib, and ribociclib, after progression (*Llombart-Cussac et al., 2025*; *Mayer et al., 2024*; *Kalinsky et al., 2025*; *Jhaveri et al., 2025*; *Kalinsky et al., 2023*). MAINTAIN and postMONARCH trials reported significantly improved progression-free survival (PFS) with continued CDK4/6i plus ET (*Kalinsky et al., 2025*). Retrospective analyses likewise suggest that second-line CDK4/6i plus ET can outperform chemotherapy or ET alone in several settings (*Martin et al., 2022*; *Ravani et al., 2025*). Safety appears comparable to or better than first-line trials, with no new toxicities and grade 3–4 events largely confined to expected hematologic adverse effects. In contrast, the PACE and PALMIRA trials did not demonstrate a PFS benefit for continuing CDK4/6i plus ET after progression (*Llombart-Cussac et al., 2025*; *Mayer et al., 2024*). Overall, the evidence for CDK4/6i continuation beyond progression is encouraging but mixed, underscoring the need for mechanistic studies of CDK4/6i-resistant proliferation to guide post-progression treatment strategies for HR⁺/HER2⁻ breast cancer.

Like CDK4/6, CDK2 also phosphorylates Rb before the G1/S transition (*Kim et al., 2022*; *Matson and Cook, 2017*). The relevance of CDK2 activation in CDK4/6i-resistant tumors highlights the potential of targeting CDK2 to overcome CDK4/6i resistance (*Pandey et al., 2020*; *Freeman-Cook et al., 2021*; *Dietrich et al., 2024*; *Al-Qasem et al., 2022*; *Kudo et al., 2024*; *Arora et al., 2023*; *Kumarasamy et al., 2025*; *Dommer et al., 2025*). Several clinical trials are currently evaluating CDK2 inhibitors (CDK2i) in patients who have progressed on CDK4/6i-based therapy, either as monotherapy or in combination with CDK4/6i, with or without ET (NCT05252416, NCT05735080). Determining the optimal strategy for incorporating CDK2i in treating HR⁺/HER2⁻ breast cancer that has developed drug resistance remains a critical question.

This study shows that, even in drug-resistant cells that remain proliferative, continued CDK4/6 inhibition results in ineffective Rb inactivation, thereby slowing E2F activation kinetics and prolonging the G1 phase. These data suggest that overall survival, rather than PFS, may serve as a more appropriate clinical trial endpoint. Moreover, our work highlights the therapeutic potential of combining CDK2i with CDK4/6i as an effective second-line strategy. Finally, we identify cyclin E overexpression as a key driver of resistance to this combination, offering mechanistic guidance for overcoming therapy resistance.

## Results

### Maintaining CDK4/6i treatment attenuates the growth of drug-resistant cells by extending G1-phase progression

To evaluate the impact of continued CDK4/6i treatment in drug-resistant settings, we employed HR⁺/HER2⁻ breast cancer cell lines (MCF-7, T47D, and CAMA-1) and a triple-negative breast cancer cell line (MDA-MB-231), all of which have an intact Rb/E2F pathway. These cells were chronically exposed to palbociclib for over a month to induce drug resistance (*Figure 1A*). We confirmed increased

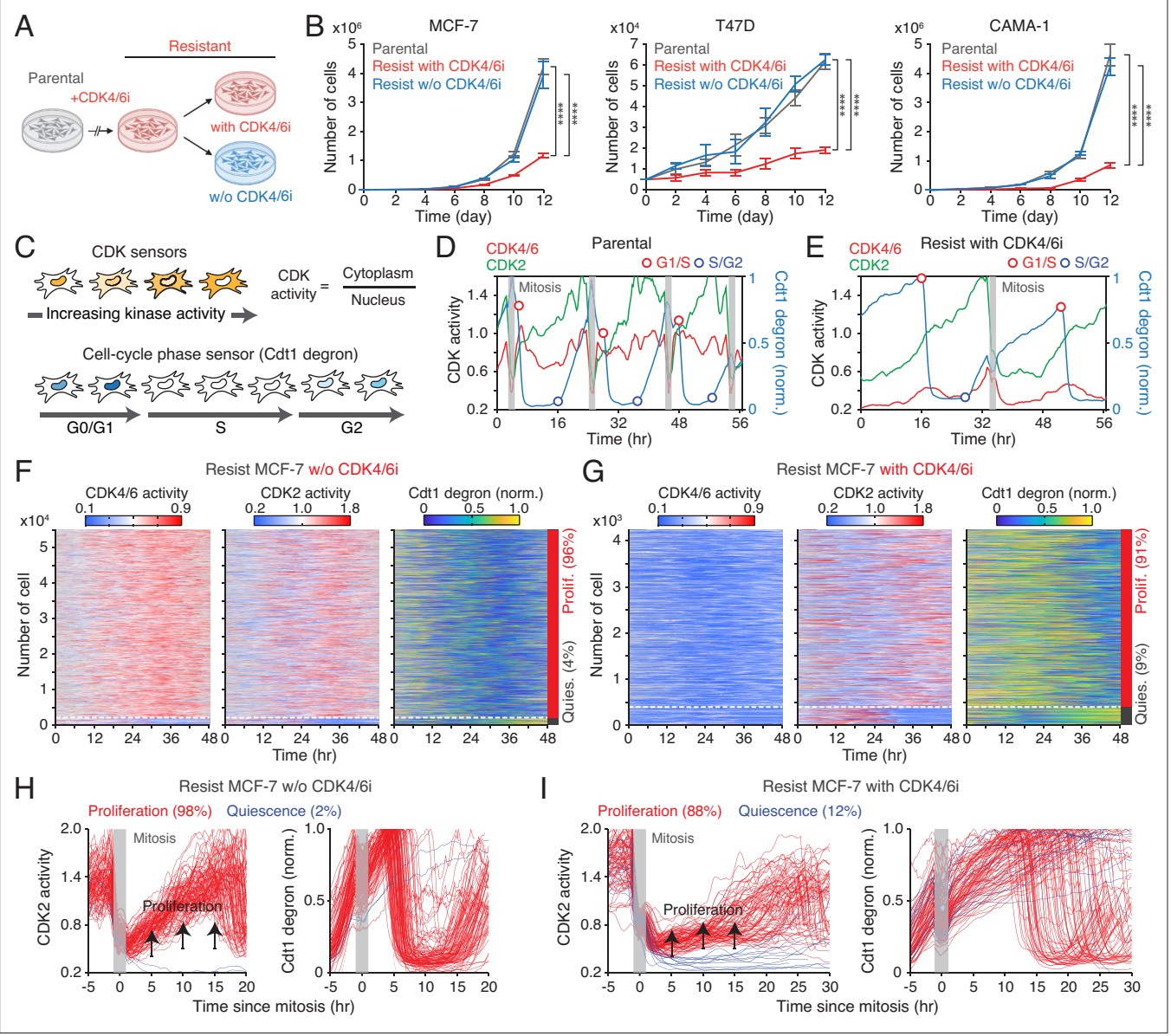

**Figure 1.** Continuous CDK4/6i treatment suppresses the growth of drug-resistant cells. (**A**) Schematic illustrating the establishment of CDK4/6i-resistant cells, either maintained in drug or withdrawn from treatment, for use in subsequent experiments. (**B**) Growth curves of parental and drug-resistant cells. Palbociclib (1 μM) was either withdrawn or maintained in drug-resistant cells. Data represent mean ± SD (n=3 biological replicates). Statistical significance was determined by two-way ANOVA with Tukey's post hoc analysis (****p<0.0001). (**C**) Schematic of live-cell reporters for CDK4/6 and CDK2 activities (top) and cell-cycle phase (bottom). (**D, E**) Representative single-cell traces showing CDK4/6 and CDK2 activities together with Cdt1-degron intensity in parental (**D**) and drug-resistant (**E**) MCF-7 cells. Resistant cells were maintained in palbociclib (1 μM). (**F, G**) Heatmaps of single-cell traces showing CDK4/6 (left) and CDK2 (middle) activities, and Cdt1-degron intensity (right) in drug-resistant cells without (**F**) or with (**G**) continuous palbociclib (1 μM). Proliferating cells were identified by sustained CDK2 activity (>1 for >2 hr during 30–48 hr). (**H, I**) Single-cell traces of CDK2 activity (left) and Cdt1-degron intensity (right), aligned to the end of mitosis (anaphase) in drug-resistant cells without (**H**) or with (**I**) continuous palbociclib (1 μM). Based on CDK2 activity (black line), cells were classified as proliferating (red) or quiescent (blue).

The online version of this article includes the following source data and figure supplement(s) for figure 1:

**Figure supplement 1.** Validation of drug resistance and visualization of cell-cycle progression.

**Figure supplement 2.** Slow cell-cycle progression in drug-resistant cells continuously treated with CDK4/6i.

**Figure supplement 3.** Potential noncanonical role of CDK6 in promoting CDK4/6i resistance.

**Figure supplement 3—source data 1.** PDF file containing original western blot for *Figure 1—figure supplement 3*, indicating the relevant bands and conditions.

**Figure supplement 3—source data 2.** Original files for western blot displayed in *Figure 1—figure supplement 3*.

half-maximal inhibitory concentrations (IC50) for palbociclib in drug-resistant cells compared to parental cells (*Figure 1—figure supplement 1A*). We maintained or withdrew CDK4/6i treatment in drug-resistant cells and monitored their cumulative proliferation alongside that of parental cells. Drug-resistant cells maintained on CDK4/6i exhibited significantly slower growth than those under other conditions (*Figure 1B*). In contrast, resistant cells withdrawn from treatment grew at rates comparable to parental cells. Thus, despite the emergence of resistance, continued CDK4/6i treatment markedly suppresses the growth rate of drug-resistant cells.

To investigate how continued CDK4/6i treatment impedes the growth of drug-resistant cells, we monitored cell-cycle progression at the single-cell level. We established MCF-7 and MDA-MB-231 cells expressing live-cell sensors for CDK4/6 activity (*Yang et al., 2020*), CDK2 activity (*Spencer et al., 2013*), and cell-cycle phase (Cdt1 degron) (*Sakaue-Sawano et al., 2017*), as well as a nucleus marker (histone 2B) for individual cell tracking. The CDK sensors dynamically shuttle between the nucleus and cytoplasm depending on phosphorylation by their respective kinases (*Figure 1C*, top). The Cdt1 degron is degraded during the S phase, allowing for the visualization of the G1/S and S/G2 transitions (*Figure 1C*, bottom). In parental cells, CDK4/6 was continuously activated after mitosis, followed by a gradual activation of CDK2, leading to S-phase entry (*Figure 1D*). In contrast, drug-resistant cells entered S phase through CDK2 activation with minimal CDK4/6 activity (*Figure 1E*). When evaluating thousands of cells, about 98% of parental cells exhibited sustained CDK4/6 and CDK2 activation to enter the cell cycle (*Figure 1—figure supplement 1C and D*). While drug withdrawal reactivated CDK4/6, over 90% of drug-resistant cells continued to proliferate regardless of the presence of CDK4/6i (*Figure 1F and G*, *Figure 1—figure supplement 1E and F*). To further assess cell-cycle dynamics, we aligned cells at cytokinesis and classified them as proliferating or quiescent based on CDK2 activity. In the absence of CDK4/6i, drug-resistant cells displayed CDK4/6 and CDK2 activity dynamics and S-phase entry kinetics comparable to those of parental cells (*Figure 1H*, *Figure 1—figure supplement*

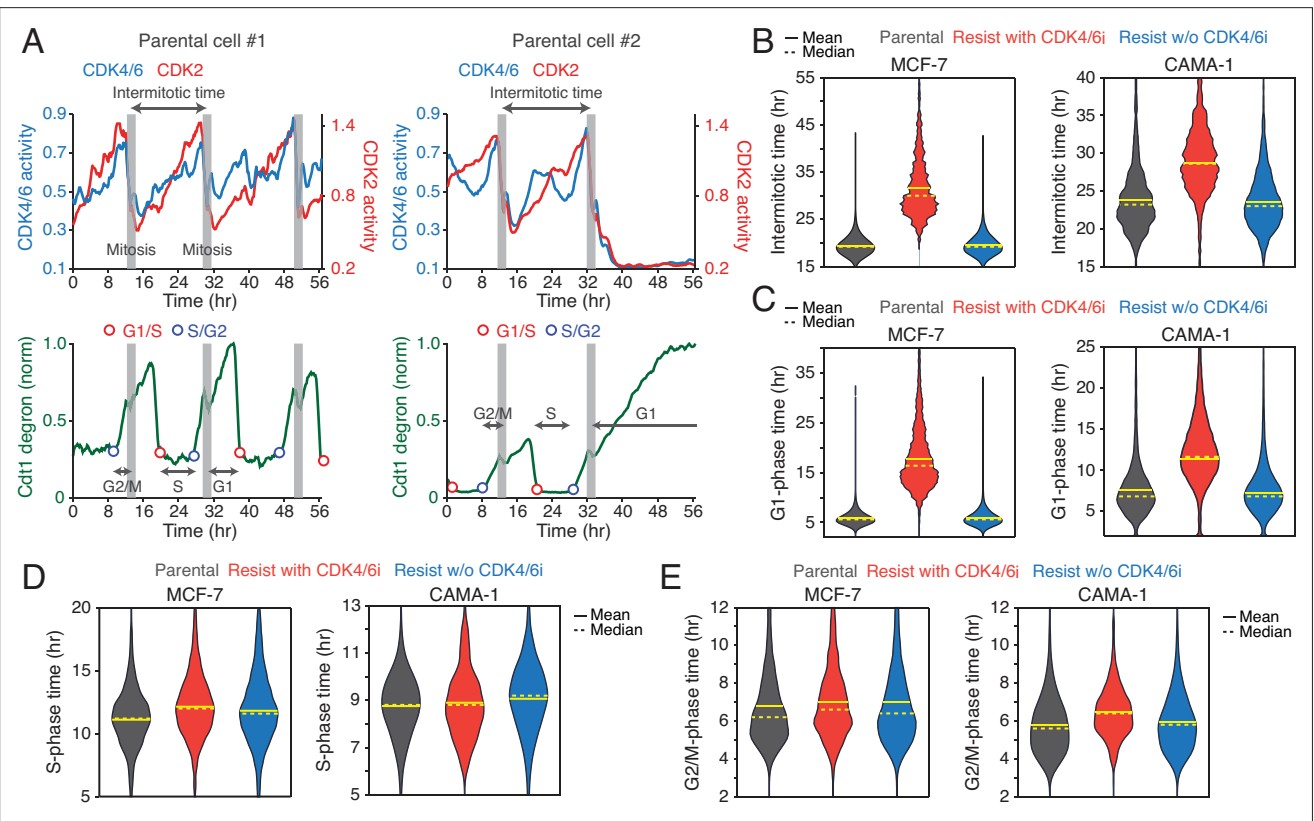

**Figure 2.** CDK4/6i maintenance extends G1-phase progression. (**A**) Representative single-cell traces showing CDK4/6 and CDK2 activities (top) and Cdt1-degron intensity (bottom) in parental MCF-7 cells. (**B–E**) Violin plots showing intermitotic time (n>600 cells/condition) (**B**), G1-phase duration (n>2000 cells/condition) (**C**), S-phase duration (n>400 cells/condition) (**D**), and G2/M-phase duration (n>250 cells/condition) (**E**) in MCF-7 (left) and CAMA-1 (right) cells. Solid and dashed yellow lines indicate mean and median, respectively.

2A–D). By contrast, continued CDK4/6i treatment in resistant cells resulted in persistently low CDK4/6 activity, delayed CDK2 activation kinetics, and greater heterogeneity in the timing of S-phase entry, as indicated by Cdt1 degradation (*Figure 1I*, *Figure 1—figure supplement 2E and F*). These results demonstrate that despite active proliferation, continuous CDK4/6i treatment markedly decelerates and destabilizes cell-cycle progression in resistant cells, thereby limiting their growth rates.

CDK6 overexpression has emerged as a key driver of CDK4/6i resistance (*Yang et al., 2017b*; *Li et al., 2022*; *Ji et al., 2020*; *Wu et al., 2021*). In line with these findings, we observed elevated CDK6 levels in CDK4/6i-resistant cells (*Figure 1—figure supplement 3A*). To investigate the role of CDK6 in modulating CDK4/6 activity under resistance, we established CDK6 knockout (KO) MCF-7 cells using CRISPR-Cas9 and induced resistance to CDK4/6i (*Figure 1—figure supplement 3B*). Despite CDK6 depletion, CDK4/6 activity remained comparable between wild-type and CDK6-KO cells under continuous CDK4/6i treatment (*Figure 1—figure supplement 3C–F*). These results suggest a potential noncanonical role for CDK6 in mediating resistance to CDK4/6i therapy.

To assess the impact of continuous CDK4/6i treatment on cell-cycle progression, we measured the intermitotic time and the duration of each cell-cycle phase in MCF-7 and CAMA-1 cells (*Figure 2A*). The intermitotic time was comparable between parental and drug-resistant cells without ongoing CDK4/6i treatment (*Figure 2B*). However, in drug-resistant cells that were continuously treated with CDK4/6i, the intermitotic time was extended by 30–50% compared to other conditions. Moreover, continuous CDK4/6i treatment prolonged the G1 phase by approximately 200–300% without affecting the duration of the S and G2/M phases (*Figure 2C–E*). Our data indicate that maintaining CDK4/6i treatment in drug-resistant cells significantly extends the G1 phase, thereby decelerating the overall growth rate.

## Maintaining CDK4/6i treatment triggers ineffective Rb inactivation

We sought to elucidate the molecular mechanisms underlying the extended G1 phase and slow CDK2 activation kinetics in drug-resistant cells under continuous CDK4/6i treatment. Our recent studies demonstrated that CDK4/6 inhibition initially halts proliferation via Rb activation but ultimately reduces Rb protein due to decreased stability (*Kim et al., 2023a*; *Zhang et al., 2023*; *Kim et al., 2025b*). Consistent with this, we found that drug-resistant cells maintained on CDK4/6i exhibited reduced levels of both total and phosphorylated Rb compared to parental cells (*Figure 3A*). Importantly, CDK4/6i withdrawal restored both total and phosphorylated Rb levels, confirming that the effect was reversible. However, Rb loss in resistant cells was incomplete, as demonstrated by comparisons with Rb-KO cells (*Figure 3—figure supplement 1A*). To directly test the role of Rb, we used Rb-KO cells resistant to CDK4/6i. Remarkably, Rb KO rescued the prolonged G1 duration and restored rapid CDK2 activation kinetics without affecting CDK4/6 activity, even under continuous CDK4/6 inhibition (*Figure 3—figure supplement 1B and C*). These findings led us to hypothesize that maintaining CDK4/6i induces a noncanonical pathway of Rb inactivation via posttranslational degradation. This passive inactivation may result in inefficient E2F activation and delayed CDK2 engagement, thereby slowing G1-phase progression.

To test this hypothesis, we tracked CDK2 activity in individual cells and aligned proliferative events with fixed-cell measurements from the end of mitosis (*Figure 3B*, *Figure 3—figure supplement 2A*). Using 5-ethynyl-2'-deoxyuridine (EdU) incorporation, we quantified the kinetics of S-phase entry. Parental and drug-resistant cells without CDK4/6i transitioned sharply into S phase approximately 6 hr after mitosis and completed DNA replication within 12 hr (*Figure 3C and D*, *Figure 3—figure supplement 2B*). In contrast, drug-resistant cells under sustained CDK4/6 inhibition exhibited markedly delayed and heterogeneous G1/S transitions (*Figure 3D*, *Figure 3—figure supplement 2C*). Quiescent cells, as expected, showed no EdU incorporation (*Figure 3E*). We next assessed the dynamics of Rb phosphorylation and E2F activity using immunostaining and mRNA FISH (*Figure 3F*). Rb hyperphosphorylation was classified by phosphorylation at Ser807/811, a well-characterized marker (*Chung et al., 2019*; *Figure 3G*). E2F1 mRNA levels, which are autoregulated (*Johnson et al., 1994*), serve as a proxy for E2F transcriptional output (*Kim et al., 2022*; *Chung et al., 2019*; *Yang et al., 2017a*). In the absence of CDK4/6i, both parental and drug-resistant cells robustly induced Rb phosphorylation before S-phase entry (*Figure 3H*). Conversely, under continuous CDK4/6 inhibition, resistant cells initiated CDK2 activation and G1 progression without Rb hyperphosphorylation, resulting in slower and weaker E2F1 mRNA induction (*Figure 3H and I*). When comparing CDK2 activity with E2F1

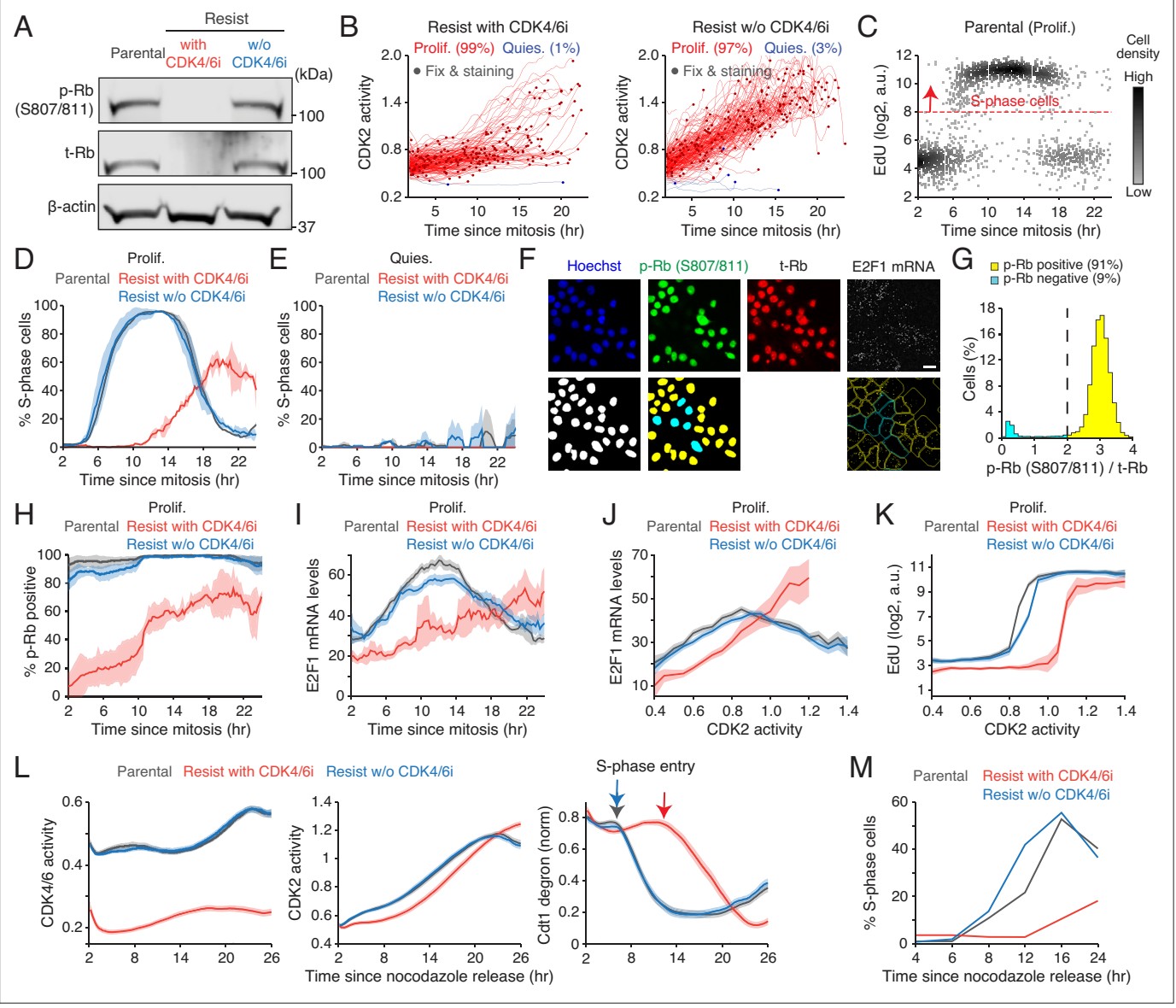

**Figure 3.** CDK4/6i maintenance induces an ineffective Rb inactivation pathway. (**A**) Immunoblot of phosphorylated Rb (S807/811; p-Rb), total Rb (t-Rb), and β-actin protein levels in MCF-7 cells. Drug-resistant cells were harvested 2 weeks after drug withdrawal. (**B**) Single-cell traces of CDK2 activity aligned to mitosis in proliferating (red) and quiescent (blue) MCF-7 cells. Circles indicate the time of fixation and staining (n=200 cells). (**C**) Scatter plot of 5-ethynyl-2'-deoxyuridine (EdU) intensity versus time since mitosis in parental cells. The red dotted line marks the threshold for S phase (n=2000 cells). (**D, E**) Percentage of S-phase cells as a function of time since mitosis in proliferating (**D**) and quiescent (**E**) cells. Data represent mean ± SD (n=2 biological replicates). (**F**) Representative immunostaining of Hoechst, p-Rb, t-Rb, and E2F1 mRNA FISH (top). Processed images showing nucleus segmentation, p-Rb classification, and mRNA puncta detection (bottom). Scale bar represents 20 μm. (**G**) Histogram of cells with p-Rb normalized to t-Rb. (**H, I**) Percentage of cells with p-Rb (**H**) and E2F1 mRNA levels (**I**) as a function of time since mitosis. Data represent mean ± SD (H, n=3 biological replicates) or mean±95% confidence intervals (I, n>2500 cells/condition). (**J, K**) E2F1 mRNA (**J**) and EdU (**K**) levels as a function of CDK2 activity. Data represent mean±95% confidence intervals (n>2500 cells/condition). (**L, M**) Averaged traces of CDK4/6 (left) and CDK2 (middle) activities and Cdt1-degron intensity (right) (**L**) and percentage of S-phase cells (**M**). MCF-7 cells were released after synchronizing with nocodazole (250 nM, 14 hr).

The online version of this article includes the following source data and figure supplement(s) for figure 3:

**Source data 1.** PDF file containing original western blot for Figure 3, indicating the relevant bands and conditions.

**Source data 2.** Original files for western blot displayed in *Figure 3*.

**Figure supplement 1.** Incomplete Rb loss mediates the extended G1-phase progression.

**Figure supplement 1—source data 1.** PDF file containing original wstern blot for *Figure 3—figure supplement 1*, indicating the relevant bands and conditions.

*Figure 3 continued on next page*

*Figure 3 continued*

**Figure supplement 1—source data 2.** Original files for western blot displayed in *Figure 3—figure supplement 1*.

**Figure supplement 2.** Slow and heterogeneous G1/S transition in drug-resistant cells maintained with CDK4/6i treatment.

mRNA levels or EdU incorporation, resistant cells under CDK4/6i maintenance required higher CDK2 activity to reach equivalent E2F1 mRNA levels or S-phase entry (*Figure 3J and K*, *Figure 3—figure supplement 2D and E*). The reduced E2F1 mRNA levels during S phase likely result from suppression by atypical E2Fs, E2F7, and E2F8 (*Westendorp et al., 2012*). Finally, mitotic synchronization with nocodazole confirmed that CDK4/6i maintenance prolongs G1 and delays S-phase entry in resistant cells (*Figure 3L and M*). Together, these findings indicate that continuous CDK4/6 inhibition in drug-resistant cells triggers inefficient Rb inactivation, attenuated E2F induction, and delayed CDK2-driven G1/S transition, leading to prolonged and heterogeneous G1-phase progression.

## Maintaining CDK4/6i suppresses the growth of drug-resistant tumors

We investigated the therapeutic benefits of maintaining CDK4/6i treatment after the onset of drug resistance in vivo. To this end, we established an MCF-7 xenograft model by orthotopically inoculating MCF-7 cells into the mammary fat pad of immunodeficient mice. Palbociclib was initially administered when tumors reached 100 mm³ (*Figure 4A*). Following an initial period of tumor stasis/regression of variable duration, tumors resumed growth, indicating the development of acquired resistance (*Figure 4B*). Upon regrowth to 145–155 mm³, mice were randomized to one of four secondary regimens: treatment discontinuation, continued palbociclib, or a switch to ribociclib or abemaciclib. To mirror clinical practice, ribociclib was administered at a fourfold higher dose than the other CDK4/6i. Consistent with our in vitro findings, continuation of any CDK4/6i significantly attenuated overall tumor growth relative to treatment discontinuation (*Figure 4C–E*). Additionally, we observed that

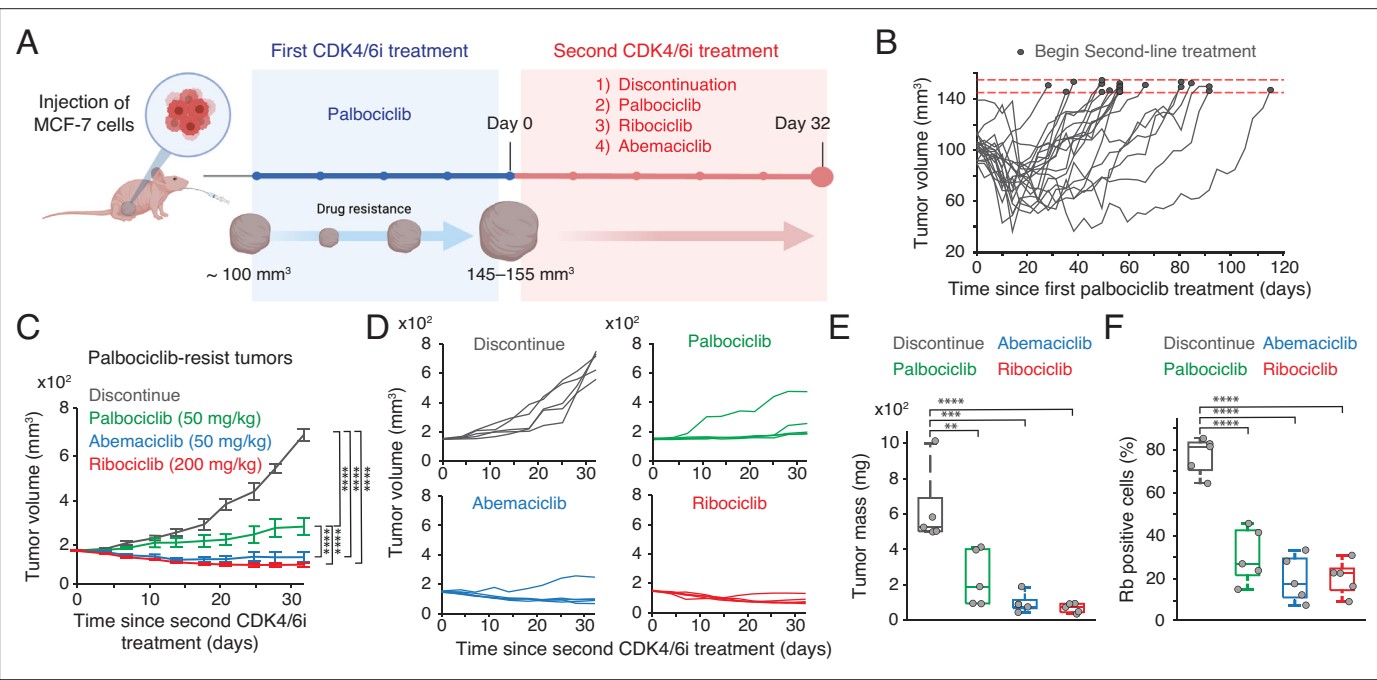

**Figure 4.** CDK4/6i maintenance suppresses the growth of drug-resistant tumors. (**A**) Schematic representation of experimental design. Once tumors reached a volume of 100 mm³, mice were treated with palbociclib. Following the development of resistance, mice were randomly assigned to one of four treatment groups: treatment discontinuation, palbociclib maintenance, switch to ribociclib, or switch to abemaciclib for 32 days. (**B**) Tumor growth curves showing the establishment of resistance to palbociclib. Horizontal red dotted lines (145–155 mm³) indicate the point at which mice were assigned to second CDK4/6i treatment (n=20 mice). (**C, D**) Averaged (**C**) and individual (**D**) tumor growth traces following second CDK4/6i treatment. Data represent mean ± SEM (n=5 mice/condition). Statistical significance was determined using two-way ANOVA with Tukey's post hoc analysis (****p<0.0001). (**E, F**) Box plots showing tumor mass (**E**) and the percentage of Rb-positive cells (**F**) after second CDK4/6i treatment (n=5 mice/condition). Statistical significance was determined using one-way ANOVA with Tukey's post hoc analysis (**p<0.05, ***p<0.01, ****p<0.0001).

switching to ribociclib or abemaciclib resulted in significantly greater tumor control compared to continuing palbociclib. These observations align with clinical studies reporting agent-specific differences in second-line PFS (*Llombart-Cussac et al., 2025*; *Mayer et al., 2024*; *Kalinsky et al., 2025*; *Kalinsky et al., 2023*). Notably, analysis of tumor tissues after secondary treatment showed restoration of total Rb expression only in the drug-discontinued cohort (*Figure 4F*). Together, these data support maintaining CDK4/6 inhibition to restrain the overall growth of drug-resistant tumors.

## Addition of ET augments the efficacy of CDK4/6i maintenance

Our recent studies have highlighted the pivotal role of c-Myc in amplifying E2F activity and fostering CDK4/6i resistance following alternative Rb inactivation (*Kim et al., 2023a*; *Zhang et al., 2023*). Using a doxycycline-inducible system, we found that induction of c-Myc significantly facilitated the growth of drug-resistant cells under ongoing CDK4/6i treatment (*Figure 5—figure supplement 1A*). Furthermore, despite similar low CDK4/6 activity, c-Myc induction facilitated cell-cycle progression in CDK4/6i-resistant cells (*Figure 5—figure supplement 1B–D*). These data demonstrated that c-Myc promotes CDK4/6i resistance by facilitating cell-cycle progression in drug-resistant cells.

Given the estrogen responsiveness of c-Myc (*Shang et al., 2000*), we hypothesized that the continued addition of ET could further suppress the growth of CDK4/6i-resistant cells by downregulating c-Myc expression. We employed the estrogen receptor antagonist fulvestrant to evaluate the therapeutic benefit of maintaining the combination of CDK4/6i and ET. While MCF-7 cells exhibited primary resistance to fulvestrant, CAMA-1 and T47D cells displayed sensitivity to this drug (*Figure 5A*). We used these MCF-7 and CAMA-1 cells to examine whether primary ET resistance contributes to the benefit of maintaining CDK4/6i and ET in drug-resistant cells. To establish drug resistance, CDK4/6i-resistant cell lines were chronically treated with a combination of palbociclib and fulvestrant for over 2 months. Subsequently, these resistant cell lines were subjected to continuous treatment with either palbociclib or fulvestrant alone or in combination, and their cumulative proliferation rate was monitored. We observed a significantly slower growth rate in drug-resistant cells continuously exposed to the combination therapy compared to other treatment conditions, not only in CAMA-1 but also in MCF-7 cells (*Figure 5B*, *Figure 5—figure supplement 2A and B*). Continued treatment with fulvestrant, either alone or combined with palbociclib, reduced c-Myc levels compared to palbociclib maintenance alone (*Figure 5C*). Our data indicate distinct resistance mechanisms between CDK4/6i and ET. Therefore, regardless of primary resistance to ET, the continued administration of ET with CDK4/6i treatment is beneficial in suppressing the growth of drug-resistant cells by inhibiting c-Myc.

To elucidate the mechanisms underlying the attenuation of growth resulting from continued treatment with the drug combination, we assessed CDK4/6 and CDK2 activities along with cell-cycle progression (Cdt1 degron) in drug-resistant MCF-7 cells. Cells were continuously treated with palbociclib or fulvestrant alone or in combination. By classifying cells into proliferation and quiescence based on CDK2 activation, we found over 80% of cells were proliferating in all conditions (*Figure 5D and E*, *Figure 5—figure supplement 2C*). Furthermore, we observed reactivation of CDK4/6 in drug-resistant cells continuously treated with fulvestrant alone. However, when we monitored cell-cycle dynamics by aligning cells to mitosis, the maintenance of the drug combination caused slower CDK2 activation kinetics and greater heterogeneity in S-phase entry, as indicated by Cdt1 degradation, compared to single-drug treatment conditions (*Figure 5F and G*, *Figure 5—figure supplement 2D and E*). By analyzing each cell-cycle duration, we observed a significant increase in intermitotic time and G1-phase duration, while S- or G2-phase lengths remained relatively unchanged (*Figure 5H and I*, *Figure 5—figure supplement 2F and G*). This emphasizes the role of the drug combination in altering cell-cycle kinetics in the G1 phase. Our findings showed the therapeutic benefits of maintaining a combination of CDK4/6i and ET in attenuating the growth of drug-resistant cells.

## The benefit of combining CDK2i with CDK4/6i as a second-line therapy

We next evaluated the therapeutic potential of adding the CDK2i INX-315 as a second-line strategy. Palbociclib/fulvestrant-resistant MCF-7 and CAMA-1 cells were monitored under five regimens: treatment discontinuation, fulvestrant alone, fulvestrant+palbociclib, fulvestrant+INX-315, or fulvestrant+palbociclib+INX-315. Continuous palbociclib+fulvestrant treatment significantly reduced resistant-cell growth compared to discontinuation, fulvestrant alone, or fulvestrant+INX-315 (*Figure 6A–C*). Notably, the triple combination provided the most robust suppression of growth. To

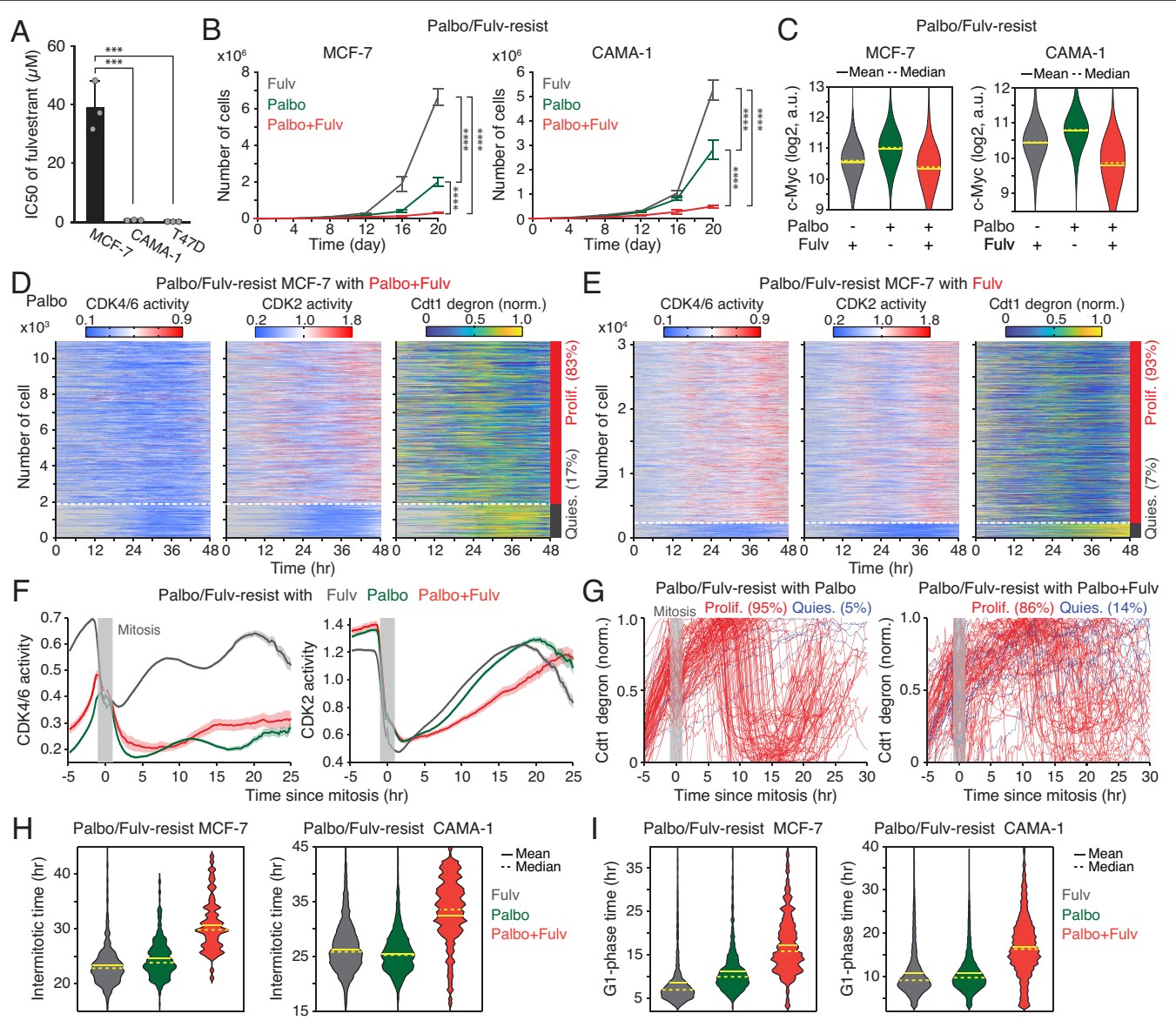

**Figure 5.** Maintaining CDK4/6i and endocrine therapy (ET) synergistically suppresses the growth of drug-resistant cells. (**A**) IC50 values of fulvestrant. Data represent mean ± SD (n=3 biological replicates). Statistical significance was determined by one-way ANOVA with Tukey's post hoc analysis (***p<0.001). (**B**) Growth curves of MCF-7 (left) and CAMA-1 (right) cells resistant to palbociclib and fulvestrant. Cells were maintained with fulvestrant (500 nM), palbociclib (1 µM), or the combination. Data represent mean ± SD (n=3 biological replicates). Statistical significance was determined by two-way ANOVA with Tukey's post hoc analysis (****p<0.0001). (**C**) Violin plots of c-Myc levels measured by immunofluorescence in resistant cells treated with the indicated drugs for 1 week. Solid and dashed yellow lines represent mean and median, respectively (n>2000 cells/condition). (**D, E**) Heatmaps showing single-cell traces for CDK4/6 (left) and CDK2 (middle) activities and Cdt1-degron intensity (right) in drug-resistant cells maintained with palbociclib (1 µM) and fulvestrant (500 nM) (**D**) or fulvestrant alone (**E**). Proliferating cells were identified based on CDK2 activity (>1 for more than 2 hr between 30 and 48 hr). (**F**) Averaged traces of CDK4/6 (left) and CDK2 (right) activities aligned by mitosis in drug-resistant cells treated with the indicated drugs. Data represent mean±95% confidence intervals (n>1000 cells/condition). (**G**) Single-cell traces of Cdt1-degron intensity aligned by mitosis in drug-resistant cells treated with palbociclib (1 µM) (left) or palbociclib+fulvestrant (500 nM) (right). (**H, I**) Violin plots showing intermitotic time (n>70 cells/condition) (**H**), and G1-phase duration (n>630 cells/condition) (**I**) in MCF-7 (left) and CAMA-1 (right) cells. Solid and dashed yellow lines indicate mean and median, respectively.

The online version of this article includes the following figure supplement(s) for figure 5:

**Figure supplement 1.** c-Myc overexpression facilitates CDK4/6i resistance by accelerating cell-cycle progression.

**Figure supplement 2.** Maintaining CDK4/6i and endocrine therapy (ET) synergistically suppresses the growth of drug-resistant cells.

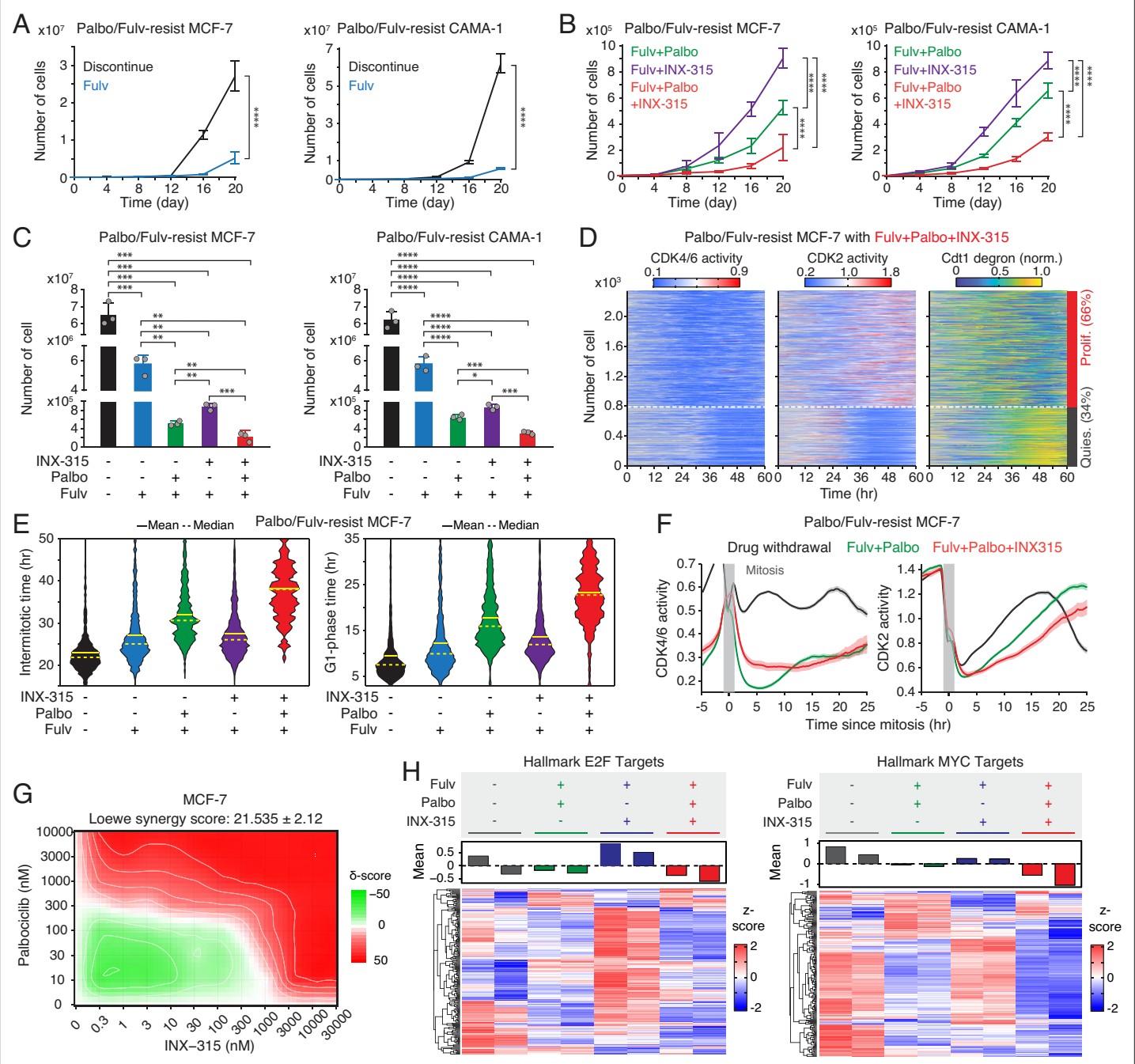

**Figure 6.** Therapeutic benefit of combining CDK2i with CDK4/6i and endocrine therapy (ET). (**A, B**) Growth curves of MCF-7 (left) and CAMA-1 (right) cells resistant to palbociclib and fulvestrant under various treatment conditions: drug discontinuation, fulvestrant (500 nM) alone, or in combination with palbociclib (1 μM) and/or INX-315 (100 nM). Data represent mean ± SD (n=3 biological replicates). p-Values were calculated using two-way ANOVA with Tukey's post hoc analysis (****p<0.0001). (**C**) Cell numbers 20 days after drug treatment. Data represent mean ± SD (n=3 biological replicates). p-Values were calculated using an unpaired *t*-test (*p<0.05, **p<0.01, ***p<0.001, ****p<0.0001). (**D**) Heatmaps of single-cell traces for CDK4/6 (left) and CDK2 (middle) activities, and Cdt1-degron intensity (right) in palbociclib/fulvestrant-resistant MCF-7 cells treated with the triple combination of palbociclib (1 μM), fulvestrant (500 nM), and INX-315 (100 nM) for 1 week before imaging. Proliferating cells were identified based on CDK2 activity (>1 for more than 2 hr between 30 and 48 hr). (**E**) Violin plots showing intermitotic time (n>200 cells/condition) (left) and G1-phase duration (n>900 cells/condition) (right). Solid and dashed yellow lines indicate mean and median, respectively. (**F**) Averaged traces of CDK4/6 (left) and CDK2 (right) activities aligned by mitosis in drug-resistant cells treated with the indicated drugs. Data represent mean±95% confidence intervals (n>1000 cells/condition). (**G**) Loewe synergy score calculated from percent-inhibition data generated by dual titration of palbociclib (0–10 μM) and INX-315 (0–30 μM) in MCF-7 cells

*Figure 6 continued on next page*

*Figure 6 continued*

following 48 hr treatment (n=3 biological replicates). (**H**) Heatmaps comparing gene expression profiles for hallmark E2F (left) and MYC (right) targets in drug-resistant MCF-7 cells treated with the indicated drugs for 20 days. Samples were collected as biological duplicates.

The online version of this article includes the following figure supplement(s) for figure 6:

**Figure supplement 1.** Combining CDK2i with CDK4/6i and endocrine therapy (ET) effectively suppresses the growth of drug-resistant cells.

dissect cell-cycle dynamics, we assessed CDK activities and Cdt1 degron levels 1 week after treatment. Consistent with earlier results, palbociclib withdrawal triggered CDK4/6 reactivation (*Figure 6—figure supplement 1A, B, and C*). Furthermore, we found CDK2 reactivation even in the presence of INX-315 (*Figure 6D*, *Figure 6—figure supplement 1D*). While CDK2 reactivation was also observed, the triple combination produced the longest intermitotic intervals and G1-phase duration without affecting the S and G2 phases (*Figure 6E*, *Figure 6—figure supplement 1E*). Analysis of cells synchronized by mitosis further confirmed that the triple-drug regimen most strongly delayed CDK2 activation kinetics (*Figure 6F*). Moreover, dose titration studies revealed a synergistic interaction between CDK4/6i and CDK2i (*Figure 6G*, *Figure 6—figure supplement 1F*). To explore transcriptomic consequences, we performed RNA sequencing 20 days post-treatment. Continued palbociclib+fulvestrant suppressed E2F target genes more effectively than INX-315, and the triple combination achieved the most potent inhibition of both E2F and Myc target gene programs (*Figure 6H*, *Figure 6—figure supplement 1G*). Together, these findings demonstrate that, despite partial CDK2 reactivation, adding CDK2i to ongoing CDK4/6i+ET further attenuates E2F activity, delays CDK2 engagement, and suppresses resistant-cell growth. These results support a therapeutic rationale for maintaining CDK4/6 blockade while strategically incorporating CDK2 inhibition after progression.

## Role of cyclin E/A in CDK2 reactivation under combined CDK4/6 and CDK2 inhibition

A recent study identifies the cyclin A-CDK complex as a key driver of rapid adaptation to CDK2 inhibition (*Arora et al., 2023*). To investigate the mechanism of CDK2 reactivation during CDK2i treatment, we explored the role of the CDK2 activators, cyclins E and A. Using a doxycycline-inducible system, we induced the overexpression of cyclin E1 or A2 in drug-naïve MCF-7 cells (*Figure 7—figure supplement 1A*). While treatment with INX-315 (100 nM to 1 µM) initially inhibited CDK2 activity, CDK2 reactivation occurred within 1–2 hr after drug treatment, regardless of cyclin E1 or A2 overexpression (*Figure 7A*, *Figure 7—figure supplement 1B–D*). The combination of CDK2i and CDK4/6i blocked this rapid CDK2 reactivation, but the overexpression of cyclin E1 or A2 hindered complete suppression (*Figure 7B*, *Figure 7—figure supplement 1E–G*). These findings suggest that CDK4/6 inhibition is necessary for effectively suppressing CDK2 activity by CDK2i, and that cyclin E1 and A2 overexpression may contribute to CDK2 reactivation.

We next examined the impact of cyclin E and A overexpression in the development of persister cells, which are implicated in residual tumor growth and eventual resistance (*Kim et al., 2023a*; *Dhanyamraju et al., 2022*). We used drug-naïve MCF-7 cells expressing live-cell sensors for CDK2, CDK4/6, and anaphase-promoting complex/cyclosome (APC/C) (Geminin-degron) activities (*Sakaue-Sawano et al., 2008*). APC/C is a multi-subunit E3 ubiquitin ligase typically inactivated near the G1/S transition, leading to Geminin-degron accumulation. Treatment with CDK4/6i and CDK2i induced near-complete cell-cycle arrest within 24 hr (*Figure 7C*). However, approximately 10% of cells developed a persister phenotype through CDK2 reactivation. Overexpression of cyclin E1 increased the percentage of persister cells, while cyclin A2 had no obvious impact (*Figure 7D*, *Figure 7—figure supplement 2*). Further kinetic analysis revealed that cyclin E1 overexpression accelerated CDK2 reactivation and APC/C inactivation in persister cells (*Figure 7E*). These findings indicate that cyclin E1, but not cyclin A2, facilitates CDK2 reactivation and persister development, promoting resistance to the CDK4/6i and CDK2i combination.

Since cyclin A is a substrate of APC/C (*Peters, 2006*), the lack of effect from cyclin A2 induction could be due to its degradation by APC/C. Thus, this degradation may reduce cyclin A to insufficient levels, preventing its impact on persister cell development. To further evaluate the impact of cyclin E and A overexpression on resistance development, we exposed palbociclib-resistant MCF-7 cells to the combination of palbociclib and INX-315. Both cyclin E1 and A2 overexpression significantly

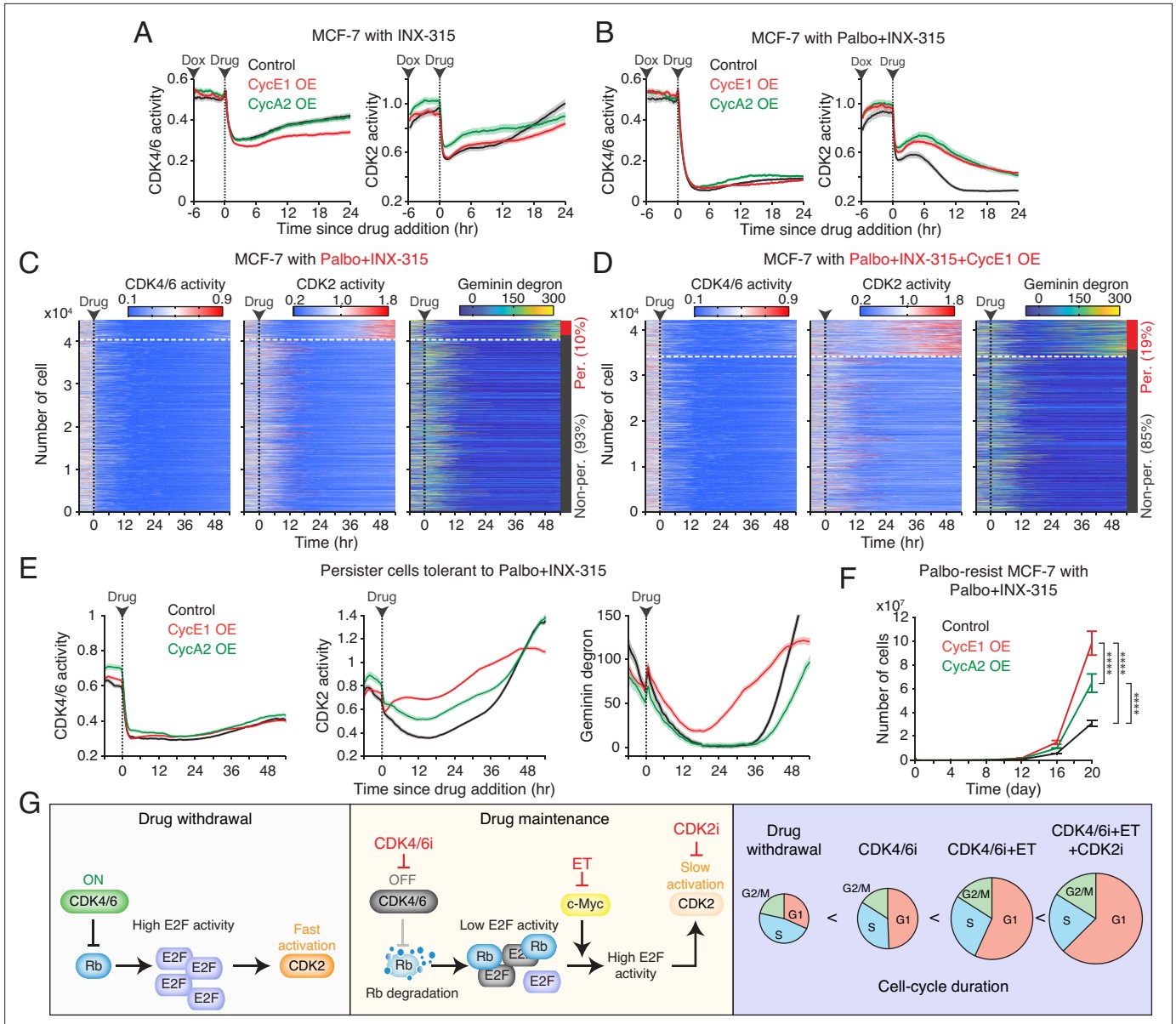

**Figure 7.** Overexpression of cyclins E and A facilitates resistance to the combination of CDK2i and CDK4/6i. (**A, B**) Averaged traces of CDK4/6 (left) and CDK2 (right) activities in MCF-7 cells with or without cyclin E1 or A2 overexpression. Cells were treated with doxycycline (500 nM) 6 hr before the addition of INX-315 (1 µM) alone (**A**) or in combination with palbociclib (1 µM) (**B**). Data represent mean ± 95% confidence intervals (n>900 cells/ condition). (**C, D**) Heatmaps of single-cell traces for CDK4/6 (left) and CDK2 (middle) activities, and Geminin-degron intensity (right) in drug-naïve MCF-7 cells without (**C**) or with (**D**) cyclin E1 overexpression. Cells were treated with palbociclib (1 µM), INX-315 (100 nM), and doxycycline (500 nM). Persister cells were identified based on CDK2 activity (>1 for more than 2 hr between 30 and 48 hr). (**E**) Averaged traces of CDK4/6 (left) and CDK2 (middle) activities and Geminin-degron intensity (right) in persister cells tolerant to combination palbociclib (1 µM), and INX-315 (100 nM) without or with cyclin E1 or A2 overexpression. Data represent mean ± 95% confidence interval. n>2500 cells/condition. (**F**) Growth curves of palbociclib-resistant MCF-7 cells without or with cyclin E1 or A2 overexpression. Cells were treated with palbociclib (1 µM), INX-315 (100 nM), and doxycycline (500 nM). Data represent mean ± SD (n=3 biological replicates). Statistical significance was determined using two-way ANOVA with Tukey's post hoc analysis (****p<0.0001). (**G**) Summary schematic illustrating the mechanisms underlying the benefits of continued CDK4/6i and endocrine therapy (ET) therapies and the introduction of CDK2i in drug-resistant cells.

The online version of this article includes the following source data and figure supplement(s) for figure 7:

**Figure supplement 1.** Cyclin E or A overexpression attenuates full CDK2 inhibition by CDK4/6i and CDK2i combination.

**Figure supplement 1—source data 1.** PDF containing original western blot for *Figure 7—figure supplement 1*, indicating the relevant bands and conditions.

*Figure 7 continued on next page*

*Figure 7 continued*

**Figure supplement 1—source data 2.** Original files for western blot displayed in *Figure 7—figure supplement 1*.

**Figure supplement 2.** Cyclin E overexpression facilitates adaptation to CDK4/6i and CDK2i combination.

accelerated the development of resistance to the drug combination, with cyclin E1 having a more significant effect than cyclin A2 (*Figure 7F*). These results indicate that although CDK2 activity is targeted by CDK2i, the overexpression of cyclin E and A promotes resistance to the combination of CDK4/6i and CDK2i. Due to cyclin A degradation by APC/C in quiescent cells, cyclin E may play a more critical role in driving resistance to this drug combination.

## Discussion

The combination of CDK4/6i and ET has reshaped treatment for HR+/HER2– breast cancer (*Johnston et al., 2019*; *Finn et al., 2015*; *Finn et al., 2016*; *Turner et al., 2018*; *Dickler et al., 2017*; *Hortobagyi et al., 2022*; *Slamon et al., 2020*; *Im et al., 2019*). However, resistance commonly emerges, and no consensus second-line standard is established. Our data show that continued CDK4/6i treatment in drug-resistant cells engages a noncanonical, proteolysis-driven route of Rb inactivation, yielding attenuated E2F output and a pronounced delay in G1 progression (*Figure 7G*). Concurrent ET further deepens this blockade by suppressing c-Myc-mediated E2F amplification, thereby prolonging G1 and slowing population growth. Importantly, CDK2 inhibition alone was insufficient to control resistant cells. Robust suppression of both CDK2 activity and resistant-cell growth required CDK2i in combination with CDK4/6i, consistent with prior reports supporting dual CDK targeting (*Pandey et al., 2020*; *Freeman-Cook et al., 2021*; *Dietrich et al., 2024*; *Al-Qasem et al., 2022*; *Kudo et al., 2024*; *Arora et al., 2023*; *Kumarasamy et al., 2025*; *Dommer et al., 2025*). Moreover, cyclin E blunted the efficacy of the CDK4/6i+CDK2i combination by reactivating CDK2. Together, these findings provide a mechanistic rationale for maintaining CDK4/6i beyond progression and support testing the combination of CDK4/6i and CDK2i, as evidenced by concordant in vitro and in vivo results.

Our data indicate that maintaining both CDK4/6i and ET synergistically decelerates cell-cycle progression in drug-resistant cells by further delaying CDK2 activation kinetics and the G1/S transition without affecting the S and G2 phases. This dual effect stems from CDK4/6i causing suboptimal Rb inactivation, while ET suppresses the global transcription amplifier c-Myc, collectively leading to diminished E2F transcriptional activity. As a result, this reduced E2F activity lowers the expression of critical cell-cycle genes, such as cyclin E and A, extending the time needed for CDK2 activation. Given that CDK2 plays an essential role in initiating and advancing DNA replication (*Tanaka et al., 2007*; *Krude et al., 1997*), its delayed activation significantly prolongs the G1/S transition. Moreover, CDK2 activation also contributes to Rb phosphorylation and inactivation. High CDK2 activity is required to phosphorylate Rb, and CDK2-mediated Rb phosphorylation is tightly coupled with DNA replication timing (*Kim et al., 2022*; *Chung et al., 2019*). Thus, upon Rb phosphorylation by CDK2 at the G1/S transition, drug-resistant cells may effectively proceed through the cell cycle even under continued CDK4/6i treatment.

Clinical trials evaluating the efficacy of sustained CDK4/6i therapy predominantly use PFS as the primary endpoint (*Llombart-Cussac et al., 2025*; *Mayer et al., 2024*; *Kalinsky et al., 2025*; *Jhaveri et al., 2025*; *Kalinsky et al., 2023*). However, our findings suggest that drug-resistant tumors continue to proliferate despite CDK4/6i maintenance. Consequently, maintaining CDK4/6i appears to slow tumor growth rather than completely arrest it. This underscores the need for clinical trials to consider overall survival and tumor progression rates as more appropriate endpoints for assessing the true benefits of sustained CDK4/6i therapy. Furthermore, the distinct polypharmacology profiles among CDK4/6i (*Hafner et al., 2019*), with ribociclib being the most specific and abemaciclib the least, may explain the varying therapeutic outcomes observed among these inhibitors (*Kalinsky et al., 2023*; *Navarro-Yepes et al., 2023*).

Maintaining CDK4/6i treatment beyond progression may be particularly beneficial for about 70% of patients who do not acquire new genetic mutations (*O'Leary et al., 2018*). However, it is important to recognize that resistance to CDK4/6i often arises from mutations in genes associated with mitogenic or hormone-signaling pathways (*O'Leary et al., 2018*; *Mao et al., 2020*; *Wander et al., 2020*; *Formisano et al., 2017*; *Costa et al., 2020*). These include mutations in *PIK3CA*, *ESR1*, *FGFR1–3*, and

*HER2*, which have been linked to increased c-Myc expression (*Shang et al., 2000*; *Zhu et al., 2008*; *Tsai et al., 2012*). Additionally, previous studies have identified *FAT1* mutations as a driver of CDK4/6i resistance (*Li et al., 2022*; *Li et al., 2018*). These resistance mutations may reduce the efficacy of maintaining CDK4/6i and ET therapy. Moreover, about 4.7% of HR⁺/HER2⁻ breast cancer patients exhibit *Rb* mutations (*O'Leary et al., 2018*; *Wander et al., 2020*), making CDK4/6i treatment unlikely to be effective, thus making its continuation inadvisable in these cases.

In conclusion, our study provides mechanistic rationale for maintaining CDK4/6i together with ET after disease progression in HR⁺/HER2⁻ breast cancers that retain an intact Rb/E2F pathway. The combination of CDK4/6i and CDK2i can further provide durable growth suppression, consistent with prior studies (*Pandey et al., 2020*; *Freeman-Cook et al., 2021*; *Dietrich et al., 2024*; *Al-Qasem et al., 2022*; *Kudo et al., 2024*; *Arora et al., 2023*; *Kumarasamy et al., 2025*; *Dommer et al., 2025*). However, it is essential to acknowledge that CDK2/4/6 inhibition may promote whole-genome duplication (*Kim et al., 2025a*), potentially fueling more aggressive tumor evolution. Finally, we identify cyclin E overexpression as a key driver of resistance to dual CDK4/6i and CDK2i therapy, providing a basis for biomarker-guided patient selection and the development of strategies to overcome therapeutic resistance.

# Materials and methods

## Key resources table

| Reagent type (species) or resource | Designation | Source or reference | Identifiers | Additional information |
|---|---|---|---|---|
| Strain, strain background (*mouse, female*) | J:NU Foxn1⁻/⁻ 4–6 weeks | Jackson Laboratories | 007850, RRID:IMSR_JAX:007850 | |
| Cell line (*Homo sapiens*) | MCF-7 | ATCC | HTB-22, RRID:CVCL_0031 | Human breast adenocarcinoma |
| Cell line (*Homo sapiens*) | T47D | ATCC | HTB-133, RRID:CVCL_0553 | Human breast adenocarcinoma |
| Cell line (*Homo sapiens*) | CAMA-1 | ATCC | HTB-21, RRID:CVCL_1115 | Human breast adenocarcinoma |
| Cell line (*Homo sapiens*) | MDA-MB-231 | ATCC CVCL_0062 | CRM-HTB-26, RRID:CVCL_0062 | Human breast adenocarcinoma |
| Antibody | Rb (mouse monoclonal) | Cell Signaling Technology | 9309,RRID:AB_823629 | 1:1000 |
| Antibody | Phospho-Rb (rabbit monoclonal) | Cell Signaling Technology | 8516, RRID:AB_11178658 | 1:1000 |
| Antibody | c-Myc (rabbit monoclonal) | Cell Signaling Technology | 5605, RRID:AB_1903938 | 1:1000 |
| Antibody | CDK6 (rabbit monoclonal) | Abcam | Ab124821, RRID:AB_10999714 | 1:1000 |
| Antibody | Cyclin E (rabbit monoclonal) | Abcam | Ab32103, RRID:AB_731789 | 1:1000 |
| Antibody | Cyclin A (rabbit monoclonal) | Abcam | Ab32386, RRID:AB_2244193 | 1:1000 |
| Recombinant DNA reagent | H2B-iRFP670-p2a-mCerulean-Cdt1 (1–100) (Plasmid) | Addgene | 223965 | |
| Recombinant DNA reagent | H2B-iRFP670-p2a-mCerulean-Geminin (1–110) (Plasmid) | Addgene | 223959 | |
| Recombinant DNA reagent | DHB (995–1087)-mVenus-p2a-mCherry-Rb (886–928) (Plasmid) | Addgene | 126679 | |
| Recombinant DNA reagent | pCW-CyclinE1-PGK-Puro (Plasmid) | Addgene | 50661 | Edited with Gibson assembly |
| Recombinant DNA reagent | pCW-CyclinA2-PGK-Puro (Plasmid) | Addgene | 50661 | Edited with Gibson assembly |
| Recombinant DNA reagent | pCW-CyclinE1-PGK-Puro (Plasmid) | Addgene | 50661 | Edited with Gibson assembly |

*Continued on next page*

*Continued*

| Reagent type (species) or resource | Designation | Source or reference | Identifiers | Additional information |
|---|---|---|---|---|
| Sequence-based reagent | CDK6 crRNA | IDT | Hs.Cas9.CDK6.1.AA | Combined 1:1 with AC |
| Sequence-based reagent | CDK6 crRNA | IDT | Hs.Cas9.CDK6.1.AC | Combined 1:1 with AA |
| Sequence-based reagent | Rb crRNA | IDT (previous paper) | Hs.Cas9.RB1.1.AA | Combined 1:1 with AB |
| Sequence-based reagent | Rb crRNA | IDT (previous paper) | Hs.Cas9.RB1.1.AB | Combined 1:1 with AA |
| Commercial assay or kit | ViewRNA ISH cell assay kit | Thermo Fisher Scientific | QVC0001 | |
| Commercial assay or kit | GeneJet RNA Purification Kit | Thermo Fisher Scientific | K0732 | |
| Chemical compound, drug | Palbociclib | Selleck Chemicals | S1116 | In vitro |
| Chemical compound, drug | Palbociclib | MedChemExpress | HY50767 | In vivo |
| Chemical compound, drug | INX-315 | MedChemExpress | HY162001 | |
| Chemical compound, drug | Fulvestrant | Selleck Chemicals | S1191 | |
| Chemical compound, drug | Ribociclib | MedChemExpress | HY15777 | |
| Chemical compound, drug | Abemaciclib | MedChemExpress | HY16297A | |
| Chemical compound, drug | Corn oil | Spectrum Chemical | CO136 | |
| Chemical compound, drug | 5-Ethynyl-2'-deoxyuridine | Sigma-Aldrich | 900584 | |
| Chemical compound, drug | AFDye-647 picolyl azide | Click Chemistry Tools | 1300 | |
| Software, algorithm | MATLAB | | | Single-cell image analysis and visualization |
| Software, algorithm | R | | | RNA-seq data processing and visualization |
| Software, algorithm | Adobe Illustrator | | | Data visualization |

## Cell culture

MCF-7 (ATCC, HTB-22), CAMA-1 (ATCC, HTB-21), MDA-MB-231 (ATCC, CRM-HTB-26), and T47D (ATCC, HTB-133) cells were authenticated by ATCC using STR profiling. MCF-7, CAMA-1, and MDA-MB-231 cells were cultured in Dulbecco's Modified Eagle Medium (DMEM; Genesee Scientific, 25-500) supplemented with 10% fetal bovine serum (FBS; Gibco, A3160601). T47D cells were maintained in RPMI-1640 (Genesee Scientific, 25-206) with 10% FBS. All cell lines were grown at 37°C in a humidified incubator with 5% $CO_2$ and were routinely tested and confirmed negative for mycoplasma contamination.

## Drugs and chemicals

For in vitro experiments, palbociclib (Selleck Chemicals, S1116), INX-315 (MedChemExpress, HY-162001), and fulvestrant (Selleck Chemicals, S1191) were dissolved in DMSO (Sigma-Aldrich, D2438). For in vivo studies, palbociclib (MedChemExpress, HY-50767), ribociclib (MedChemExpress, HY-15777), and abemaciclib (MedChemExpress, HY-16297A) were prepared in corn oil (Spectrum Chemical, CO136) containing 10% DMSO. EdU (Sigma-Aldrich, 900584) and AFDye-647 picolyl azide (Click Chemistry Tools, 1300) were used to label DNA synthesis.

## Antibodies

Primary antibodies were from Cell Signaling Technology: Rb (9309), phospho-Rb (Ser807/811; 8516), and c-Myc (5605); from Abcam: CDK6 (ab124821), cyclin E (ab32103), and cyclin A (ab32386); and from BD Biosciences: Rb (554136). The BD Biosciences Rb antibody was used for Rb visualization in cells expressing the CDK4/6 sensor. Secondary antibodies for immunofluorescence (IF) were Alexa Fluor 488 goat anti-mouse (Thermo Fisher Scientific, A32723) and Alexa Fluor 568 goat anti-rabbit (Thermo Fisher Scientific, A11036). For immunoblotting, IRDye secondaries were used: goat anti-mouse 800CW (LI-COR, 926-32210) and goat anti-rabbit 680RD (LI-COR, 926-68071).

## DNA constructs and cell line generation

DNA constructs were generated as described previously (*Kim et al., 2022*; *Yang et al., 2020*; *Kim et al., 2023a*; *Zhang et al., 2023*; *Kim et al., 2023b*). Briefly, Gibson assembly was used to clone H2B-iRFP670-p2a-mCerulean-Cdt1 (1–100) (Addgene, 223965), H2B-iRFP670-p2a-mCerulean-Geminin (1–110) (Addgene, 223959), and DHB (995–1087)-mVenus-p2a-mCherry-Rb (886–928) (Addgene, 126679) into pLenti-IRES vectors bearing puromycin, blasticidin, or neomycin selection markers. Doxycycline-inducible c-Myc, cyclin E1, and cyclin A2 constructs were generated by inserting PCR products into pCW57.1 (Addgene, 50661) after NheI/BamHI digestion.

Stable cell lines were created by lentiviral transduction. Lentiviral plasmids and packaging vectors pMDLg/pRRE (Addgene, 12251), pRSV-Rev (Addgene, 12253), and pCMV-VSV-G (Addgene, 8454) were transfected into HEK-293T cells using polyethyleneimine in Opti-MEM (Thermo Fisher Scientific, 31985070). Viral supernatants were collected at 72 and 96 hr, pooled, clarified (1200 rpm, 5 min), filtered (0.45 µm; Millipore, SLHA033SB), concentrated (Amicon Ultra-15; Millipore, UFC910024; 4000 rpm, 10 min), and stored at −80°C. Target cells were infected in the presence of polybrene (5 µM) for 48 hr and selected with puromycin (1 µg/mL; InvivoGen, ant-pr), blasticidin (10 µg/mL; InvivoGen, ant-bl), or neomycin (800 µg/mL; Thermo Fisher Scientific, BP673-5), or by FACS based on introduced fluorescent reporters.

## CDK6 and Rb KO cell lines

Rb KO in MCF-7 cells was previously described and validated (*Kim et al., 2023a*). For CDK6 KO, MCF-7 cells were transfected with crRNA:tracrRNA-ATTO550 duplexes complexed with recombinant Cas9 to form RNPs. ATTO550-positive cells were single-cell sorted by flow cytometry, expanded, and screened by immunoblotting. Two gRNAs were used to target CDK6:

- *CDK6* gRNA 1: 5'-GACCACGUUGGGGGUGCUCGAGUUUUAGAGCUAUGCU-3'
- *CDK6* gRNA 2: 5'-CUGGACUGGAGCAAGACUUCGUUUUAGAGCUAUGCU-3'

## Drug-resistant cell lines

To generate CDK4/6i resistance, breast cancer cell lines were continuously exposed to palbociclib (1 µM) for over a month, with media and drug refreshed every 2–3 days. To establish double resistance to palbociclib and fulvestrant, palbociclib-resistant cells were then treated with fulvestrant (500 nM) for >2 months under the same refresh schedule. Resistance was confirmed by drug titration and determination of the half-maximal inhibitory concentration, calculated from the fraction of S-phase cells relative to drug-naïve controls.

## Live-cell reporters

H2B-iRFP670 was used to segment nuclei and track single cells. A fluorescent Cdt1 degron (aa 1–100) (*Sakaue-Sawano et al., 2017*) reported cell-cycle phase transitions; fluorescent Geminin was used to mark S-phase entry. Kinase translocation reporters (KTRs) for CDK4/6 and CDK2 (*Yang et al., 2020*; *Spencer et al., 2013*) monitored kinase-dependent phosphorylation of specific substrates (Rb C-terminus [aa 886–928] for CDK4/6 and DNA helicase B [DHB; aa 994–1087] for CDK2), driving regulated nucleo-cytoplasmic shuttling. The cytoplasmic/nuclear fluorescence intensity ratio provided a quantitative readout of kinase activity. Because the CDK4/6 KTR contains a degenerate CDK2 motif and partially reports CDK2 activity in S/G2, we applied a linear regression-derived correction based on the CDK2 reporter (*Yang et al., 2020*; *Kim et al., 2023a*; *Kim et al., 2025b*; *Kim et al., 2023b*; *Yang, 2024*):

- MCF-7: CDK4/6 activity = (CDK4/6 reporter) − 0.41 × (CDK2 reporter)
- MDA-MB-231: CDK4/6 activity = (CDK4/6 reporter) − 0.35 × (CDK2 reporter)

## IF and mRNA FISH

Cells were seeded in glass-bottom 96-well plates (Cellvis, P96-1.5H-N) at least 16 hr before experiments. S-phase fraction was assessed by EdU incorporation. Cells were pulsed with EdU (10 µM) for 15 min at 37°C, fixed in 4% formaldehyde (Thermo Scientific, 28906) in PBS supplemented with 10 mM HEPES (Sigma-Aldrich, H3537) for 15 min at room temperature (RT), and permeabilized in

0.2% Triton X-100 in PBS (PBS-T; Sigma-Aldrich, T8787) for 15 min. EdU was detected by click chemistry (15 min at RT) in 2 mM $CuSO_4$ (Sigma-Aldrich, C1297), 20 mg/mL sodium ascorbate (Sigma-Aldrich, A4034), and 3 µM Alexa Fluor Dye 647 picolyl azide (Click Chemistry Tool, 1300) prepared in TBS (pH 8.3; Sigma-Aldrich, T6066).

For IF, cells were blocked (1 hr, RT) in PBS containing 10% FBS, 1% BSA, 0.1% Triton X-100, and 0.01% $NaN_3$. Primary antibodies were incubated overnight at 4°C in blocking buffer: anti-phospho-Rb (Ser807/811; Cell Signaling Technology, 8516), anti-Rb (BD Biosciences, 554136), and anti-c-Myc (Cell Signaling Technology, 5605). The next day, Alexa Fluor 488- or 568-conjugated secondary antibodies (Thermo Fisher Scientific) were applied for 1 hr at RT (1:2000 in blocking buffer). Nuclei were counterstained with Hoechst 33342 (Thermo Fisher Scientific, 62249; 10 mg/mL stock diluted 1:10,000 in PBS, 15 min, RT). Cells were washed with PBS between steps and stored in PBS until imaging. For mRNA FISH, the ViewRNA ISH Cell Assay Kit (Thermo Fisher Scientific, QVC0001) and an E2F1 probe set (Thermo Fisher Scientific, VA1-12108-VC) were used according to the manufacturer's instructions.

For experiments in *Figure 3*, we used a live-fixed pipeline. Asynchronously proliferating cells were imaged live for ≥48 hr (37°C, 5% $CO_2$; acquisition parameters detailed in 'Live-cell, fixed-cell, and tumor image acquisition'). Histone H2B fluorescence was used to segment/track nuclei and define time since mitosis (t=0 at anaphase), and a CDK2 KTR provided CDK2 activity in the last live frame. Immediately after the live acquisition, the same wells were pulsed with EdU (10 µM, 15 min) and fixed/permeabilized as above. To prevent interference from fluorescent proteins during fixed assays, we photobleached residual fluorescence with 3% $H_2O_2$+20 mM HCl in PBS for 2 hr at RT. We then performed click-chemistry EdU detection, IF for phospho-Rb (Ser807/811) and total Rb, and RNA FISH for E2F1.

## Growth curve measurement

To assess population growth, 5000 cells were seeded per well in 24-well plates (Thermo Fisher Scientific, FB012929). At the indicated time points, cells were harvested by trypsinization, resuspended in 1 mL growth medium, and counted manually with a hemocytometer (Hausser Scientific, HS-3510). Cells were replated as needed for subsequent time points.

## Live-cell, fixed-cell, and tumor tissue image acquisition

Cells were maintained at 30–80% confluency in 96-well glass-bottom plates (Cellvis, P96-1.5P) (*Yang, 2024*). Phenol red-free media were used to minimize background fluorescence. Imaging was performed on an inverted Nikon Eclipse Ti-2 microscope equipped with a Hamamatsu ORCA-Fusion camera. Objectives were 10× (Nikon CFI Plan Apo Lambda, NA 0.45; no binning) or 20× (Nikon CFI Plan Apo Lambda, NA 0.75; 2×2 binning). Excitation/emission settings were: CFP, 440/482 nm; GFP, 488/525 nm; YFP, 514/535 nm; mCherry, 594/609 nm; Cy5, 640/690 nm. Images were acquired using NIS-Elements AR v5.21.03. Live-cell imaging was performed in a humidified 37°C/5% $CO_2$ chamber, acquiring 3 sites per well every 12 min. For fixed-cell imaging, 9 (10×) or 32 (20×) sites per well were collected. Total exposure time was kept <500 ms per time point. mRNA FISH images were collected with 1.5 µm z-stacks. Whole-tumor cross-sections were imaged on a Zeiss Axio Observer 7 with Apotome 2 and a Hamamatsu ORCA-Flash 4 camera using a 10× Zeiss Plan-Apochromat objective (NA 0.45; no binning) at 1.5 µm z-stack intervals.

## Image processing and analysis

Live- and fixed-cell images were processed using custom MATLAB scripts (MathWorks, R2021a). Flat-field correction was applied to reduce illumination bias. For fixed-cell analysis, nuclei were segmented on Hoechst using a histogram-curvature threshold. For live-cell analysis, nuclei were segmented using H2B-iRFP670 with a Laplacian-of-Gaussian blob detector; adjacent nuclei were separated by marker-based watershed. Background was corrected by subtracting the 50th percentile of non-nuclear pixels. For live-to-fixed alignment, cells were segmented in the final live frame and matched to the corresponding fixed-cell field. Cell tracks were generated with a deflection-bridging algorithm. Mitoses were identified by the appearance of two daughter nuclei adjacent to a prior nucleus and by H2B intensity (each daughter about 45–55% of the mother). Cytoplasmic signal was quantified as the median intensity within a ring 0.65–3.25 µm outside the nuclear boundary; cells with overlapping rings were excluded. For RNA FISH, whole-cell areas were estimated by dilating nuclear masks by 50 µm;

cells overlapping neighbors were excluded. Raw FISH images were processed with a top-hat filter (4 µm circular kernel) to generate puncta masks. Tissue images were analyzed in ImageJ.

## Immunoblot

Cells were rinsed with ice-cold PBS and lysed in 300 µL CHAPS buffer (50 mM Tris-HCl, 150 mM NaCl, 1 mM EDTA, 10 mM N-ethylmaleimide, 0.3% CHAPS, 1 mM PMSF) supplemented with Halt Protease Inhibitor (Thermo Fisher Scientific, 1861279) and phosphatase inhibitor (Roche, 4906845001) for 5 min on ice. Lysates were cleared (14,000×$g$, 10 min, 4°C). Protein concentration was determined with the Pierce 660 nm assay (Thermo Fisher Scientific, 22660). Samples (12 µg) were denatured in LDS buffer (Thermo Fisher Scientific, NP0007) at 70°C for 10 min, resolved on NuPAGE 4–12% Bis-Tris gels (Thermo Fisher Scientific, NP0322BOX), and transferred to 0.45 µm PVDF membranes (Bio-Rad, 1620174) using a Trans-Blot Turbo (Bio-Rad, 1704150) with 1× transfer buffer (Bio-Rad, 10026938). Membranes were blocked (LI-COR Intercept, 927-60001, 1 hr, RT) and incubated overnight (4°C) with primary antibodies—Rb (CST, 9309), phospho-Rb Ser807/811 (CST, 8516), CDK6 (Abcam, ab124821), or β-actin (CST, 3700S)—diluted 1:1000 in blocking buffer. After TBS-T washes (20 mM Tris, 150 mM NaCl, 0.1% Tween-20, pH 7.5), membranes were incubated with IRDye 800CW goat anti-mouse (LI-COR, 926-32210) or IRDye 680RD goat anti-rabbit (LI-COR, 926-68071) secondaries (1:2000; 2 hr, RT). Blots were imaged on a LI-COR Odyssey and processed in Image Studio Lite v5.2.

## RNA sequencing

Total RNA was extracted using the GeneJET RNA Purification Kit (Thermo Fisher Scientific, K0732) according to the manufacturer's protocol. After trypsinization and PBS wash, cells were lysed in 600 µL lysis buffer supplemented with β-mercaptoethanol (Bio-Rad, 1610710), homogenized, mixed with 360 µL 100% ethanol (Fisher Scientific, BP28184), and purified on spin columns. RNA was eluted in nuclease-free water (Thermo Fisher Scientific, 10977-015). Azenta Life Sciences performed library preparation and sequencing. Count-matrix analysis was conducted in R using DESeq2 (v1.44.0) (*Love et al., 2014*) with log$_2$ fold-change shrinkage via apeglm (*Zhu et al., 2019*). Gene sets (Hallmark, C2, C5) were obtained from MSigDB via msigdbr (v7.5.1). Heatmaps were generated with Complex-Heatmap (*Gu and Hübschmann, 2022*). GSEA was performed with clusterProfiler (v4.12.6) using Hallmark annotations, a maximum gene-set size of 500 and 10,000 permutations; gene sets with nominal p<0.05 and FDR<0.25 were considered significant.

## Animal

All procedures were approved by the Institutional Animal Care and Use Committee at Columbia University Irving Medical Center and conducted in accordance with NIH guidelines. Female J:NU Foxn1$^{-/-}$ mice (6 weeks of age; The Jackson Laboratory, 007850) were housed in a specific pathogen-free barrier facility on a 12 hr light/dark cycle with ad libitum access to food and water. Mice were monitored per institutional humane endpoint policies under IACUC animal protocol AC-AABR6602.

## Xenograft experiments

Female J:NU mice were anesthetized by intraperitoneal injection of ketamine (45 mg/kg) plus xylazine (5 mg/kg). Drug-naïve MCF-7 cells (8×10$^6$ per mouse) were resuspended 1:1 in PBS:Geltrex LDEV-Free Reduced Growth Factor Basement Membrane Matrix (Thermo Fisher Scientific, A1413201) and injected orthotopically into the abdominal mammary fat pad. Tumors were measured twice weekly with digital calipers; volumes were calculated as (width$^2$×length)×0.5. When tumors reached approximately 100 mm$^3$, mice received daily palbociclib (50 mg/kg, oral gavage). Upon regrowth to 145–155 mm$^3$, mice were randomized (n=5/group) to: (1) treatment discontinuation; (2) palbociclib 50 mg/kg daily; (3) abemaciclib 50 mg/kg daily; or (4) ribociclib 200 mg/kg daily for 32 days. At study end, mice were anesthetized and perfused with 1% formaldehyde prior to tumor collection.

## Immunohistochemistry

After vascular perfusion with 1% formaldehyde in PBS, tumors were fixed in 1% formaldehyde (1 hr, 4°C), washed in PBS-T, and cryoprotected in 30% sucrose (overnight, 4°C). Samples were embedded in OCT (Thermo Fisher Scientific, 23-730-571), frozen at −80°C, and sectioned at 40 µm onto Superfrost Plus slides (Thermo Fisher Scientific, 12-550-15). Sections were rehydrated in PBS-T, blocked in 5%

normal goat serum (Jackson ImmunoResearch, 005-000-121) for 1 hr (RT), and incubated with recombinant anti-Rb (Abcam, ab181616; 1:1000 in 5% serum/PBS-T) overnight (RT). After PBS-T washes, sections were incubated with Alexa Fluor 488 goat anti-rabbit (Thermo Fisher Scientific, A32731; 1:500 in PBS-T) for 4 hr (RT), counterstained with DAPI (Sigma-Aldrich, D9542; 1:500, 10 min), and mounted in Fluoromount-G (Invitrogen, 00-4958-02) for imaging.

## Statistics and reproducibility

Statistical analyses were performed in GraphPad Prism v10.2.0. For parametric data, unpaired two-tailed Student's t-tests or one-way ANOVA with Tukey's post hoc analysis were used as appropriate; paired two-tailed t-tests were used for paired comparisons. The specific tests and sample sizes are reported in the figure legends; a comprehensive summary is provided in *Supplementary file 1*. All experiments were reproduced in at least two independent biological replicates.

## Acknowledgements

We thank Sarat Chandarlapaty for providing the CAMA-1 cells and Kevin Gardner for offering MDA-MB-231 cells. We thank Caitlin O'Neil for assisting with tissue imaging. We thank Anusha Shanabag for her assistance with the RNA-seq data processing. This work was supported by Neuroendocrine Tumor Research Foundation (MK, 855538), Melanoma Research Foundation (MK and HY), Research Scholar Grant (MK, RSG-22-167-01-MM and HY, RSG-22-101-01-CDP), V Scholar (HY, V2023-017), NIH grants R37-CA266270 (MK), R03-AG073833 (MK), R01-GM145884 (HY), and HICCC Grant P30-CA013696 (MK and HY). These studies used the resources of HICCC Flow Cytometry Shared Resources (P30-CA013696).

## Additional information

### Funding

| Funder | Grant reference number | Author |
|---|---|---|
| Neuroendocrine Tumor Research Foundation | 855538 | Minah Kim |
| Melanoma Research Foundation | | Minah Kim Hee Won Yang |
| American Cancer Society | RSG-22-167-01-MM | Minah Kim |
| American Cancer Society | RSG-22-101-01-CDP | Hee Won Yang |
| V Foundation for Cancer Research | V2023-017 | Hee Won Yang |
| National Cancer Institute | R37-CA266270 | Minah Kim |
| National Institute on Aging | R03-AG073833 | Minah Kim |
| National Institute of General Medical Sciences | R01-GM145884 | Hee Won Yang |
| National Cancer Institute | P30-CA013696 | Minah Kim Hee Won Yang |

The funders had no role in study design, data collection and interpretation, or the decision to submit the work for publication.

### Author contributions

Jessica Armand, Conceptualization, Resources, Data curation, Formal analysis, Validation, Investigation, Visualization, Methodology, Writing – review and editing; Sungsoo Kim, Resources, Formal analysis, Investigation; Kibum Kim, Investigation; Eugene Son, Formal analysis; Minah Kim, Resources, Funding acquisition, Writing – review and editing; Kevin Kalinsky, Conceptualization; Hee Won Yang, Conceptualization, Data curation, Formal analysis, Supervision, Funding acquisition, Investigation, Visualization, Writing – original draft, Project administration, Writing – review and editing

## Author ORCIDs
Jessica Armand http://orcid.org/0000-0003-1580-4477
Hee Won Yang https://orcid.org/0000-0002-0000-8091

## Ethics
All procedures were approved by the Institutional Animal Care and Use Committee at Columbia University Irving Medical Center and conducted in accordance with NIH guidelines. Female Foxn1-/-J:NU mice (6 weeks old; The Jackson Laboratory, 007850) were housed in a specific pathogen-free barrier facility on a 12-hr light/dark cycle with ad libitum access to food and water and were monitored per institutional humane-endpoint policies. IACUC animal protocol AC-AABR6602.

Reviewer #1 (Public review): https://doi.org/10.7554/eLife.104545.3.sa1
Reviewer #2 (Public review): https://doi.org/10.7554/eLife.104545.3.sa2
Reviewer #3 (Public review): https://doi.org/10.7554/eLife.104545.3.sa3
Author response https://doi.org/10.7554/eLife.104545.3.sa4

# Additional files

## Supplementary files
Supplementary file 1. Statistics summary.

MDAR checklist

## Data availability
All data supporting the findings of this study are available within the article, the Source Data files, and Supplementary File 1. Bulk RNA-seq data have been deposited in the Gene Expression Omnibus under accession GSE279160. Imaging analysis code for live- and fixed -cell analysis was adapted from scripts used in *Kim et al., 2023a*, and can be found at https://github.com/tjdtn5160/image-analysis-kim-2023 (copy archived at *Kim, 2023*). Code used for analysis of RNA-seq data was adapted from scripts used in *Kim et al., 2023a*, and can be found at https://github.com/Kim-Yang-Lab/CDK4-6i_CDK7i_paper (copy archived at *Kim-Yang-Lab, 2024*).

The following dataset was generated:

| Author(s) | Year | Dataset title | Dataset URL | Database and Identifier |
|---|---|---|---|---|
| Armand JS, Yang H | 2024 | Therapeutic benefits of maintaining CDK4/6 inhibitors and incorporating CDK2 inhibitors beyond progression in breast cancer | https://www.ncbi.nlm.nih.gov/geo/query/acc.cgi?acc=GSE279160 | NCBI Gene Expression Omnibus, GSE279160 |

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
