## [Editor Report · eLife Assessment]

This study presents **fundamental** insights into overcoming resistance in hormone receptor-positive breast cancer by demonstrating that sustained CDK4/6 inhibitor treatment, either alone or in combination with CDK2 inhibitors, significantly suppresses the growth of drug-resistant cancer cells. The findings are supported by **compelling** evidence from both in vitro cell line experiments and in vivo mouse models, highlighting the therapeutic potential of maintaining CDK4/6 inhibitors beyond disease progression. Additionally, the identification of cyclin E overexpression as a key driver of resistance offers a target that will be of value for future therapeutic strategies, potentially improving outcomes for patients with advanced breast cancer.

---

## [Referee Report · Reviewer #1 (Public review)]

Summary:

In the research manuscript submitted to eLife (Manuscript ID eLife-RP-RA-2024-104545) titled "Therapeutic benefits of maintaining CDK4/6 inhibitors and incorporating CDK2 inhibitors beyond progression in breast cancer" authors identified (1) CDK4/6i treatment attenuates the growth of drug-resistant cell by prolongation of G1 phase; (2) CDK4/6i treatment results in an ineffective Rb inactivation pathways and suppress the growth of drug-resistant tumors; (3) Addition of endocrine therapy augments the efficacy of CDK4/6i maintenance; (4) Addition of CDK2i with CDK4/6 treatment as second-line treatment can suppress the growth of resistant cell; (5) finally role of cyclin E as key driver of resistance to CDK4/6 and CDK2 inhibition.

Strengths:

To prove authors complicated proposal, authors employed orchestration of several kinds of live cell markers, timed in situ hybridization, IF and Immono-bloting. The authors strongly recognize the resistance of CDK4/6 + ET therapy and demonstrated how to overcome it.

Weaknesses:

None.

Comments on revisions:

In response to the reviewers' questions and comments, the authors have revised the manuscript accordingly and sufficiently addressed the differences between their study and previous works on CDK4/6 and CDK2 combination therapy as a second-line approach.

---

## [Referee Report · Reviewer #2 (Public review)]

Summary:

This study elucidated the mechanism underlying drug resistance induced by CDK4/6i as a single agent and proposed a novel and efficacious second-line therapeutic strategy. It highlighted the potential of combining CDK2i with CDK4/6i for the treatment of HR+/HER2- breast cancer.

Strengths:

The study demonstrated that CDK4/6 induces drug resistance by impairing Rb activation, which results in diminished E2F activity and a delay in G1 phase progression. It suggests that the synergistic use of CDK2i and CDK4/6i may represent a promising second-line treatment approach. Addressing critical clinical challenges, this study holds substantial practical implications.

Comments on revisions:

The author has comprehensively addressed all the questions I raised.

---

## [Referee Report · Reviewer #3 (Public review)]

Summary:

In their manuscript, Armand and colleagues investigate the potential of continuing CDK4/6 inhibitors or combining them with CDK2 inhibitors in the treatment of breast cancer that has developed resistance to initial therapy. Utilizing cellular and animal models, the research examines whether maintaining CDK4/6 inhibition or adding CDK2 inhibitors can effectively control tumor growth after resistance has set in. The key findings from the study indicate that the sustained use of CDK4/6 inhibitors can slow down the proliferation of cancer cells that have become resistant, and the combination of CDK2 inhibitors with CDK4/6 inhibitors can further enhance the suppression of tumor growth. Additionally, the study identifies that high levels of Cyclin E play a significant role in resistance to the combined therapy. These results suggest that continuing CDK4/6 inhibitors along with the strategic use of CDK2 inhibitors could be an effective strategy to overcome treatment resistance in hormone receptor-positive breast cancer. However, several issues need to be addressed before considering its publication.

Strengths:

(1) Continuous CDK4/6 Inhibitor Treatment Significantly Suppresses the Growth of Drug-Resistant HR+ Breast Cancer: The study demonstrates that the continued use of CDK4/6 inhibitors, even after disease progression, can significantly inhibit the growth of drug-resistant breast cancer.

(2) Potential of Combined Use of CDK2 Inhibitors with CDK4/6 Inhibitors: The research highlights the potential of combining CDK2 inhibitors with CDK4/6 inhibitors to effectively suppress CDK2 activity and overcome drug resistance.

(3) Discovery of Cyclin E Overexpression as a Key Driver: The study identifies overexpression of cyclin E as a key driver of resistance to the combination of CDK4/6 and CDK2 inhibitors, providing insights for future cancer treatments.

(4) Consistency of In Vitro and In Vivo Experimental Results: The study obtained supportive results from both in vitro cell experiments and in vivo tumor models, enhancing the reliability of the research.

(5) Validation with Multiple Cell Lines: The research utilized multiple HR+/HER2- breast cancer cell lines (such as MCF-7, T47D, CAMA-1) and triple-negative breast cancer cell lines (such as MDA-MB-231), validating the broad applicability of the results.

Comments on revisions:

The authors made a significant effort to improve the manuscript. My comments were sufficiently addressed.

---

## [Author Response]

The following is the authors’ response to the original reviews.

**Reviewer #1 (Public review):**
Summary:In this manuscript, the authors identified that(1) CDK4/6i treatment attenuates the growth of drug-resistant cells by prolongation of the G1 phase;(2) CDK4/6i treatment results in an ineffective Rb inactivation pathway and suppresses the growth of drugresistant tumors;(3) Addition of endocrine therapy augments the efficacy of CDK4/6i maintenance;(4) Addition of CDK2i with CDK4/6 treatment as second-line treatment can suppress the growth of resistant cell;(5) The role of cyclin E as a key driver of resistance to CDK4/6 and CDK2 inhibition.Strengths:To prove their complicated proposal, the authors employed orchestration of several kinds of live cell markers, timed in situ hybridization, IF and Immunoblotting. The authors strongly recognize the resistance of CDK4/6 + ET therapy and demonstrated how to overcome it.Weaknesses:The authors need to underscore their proposed results from what is to be achieved by them and by other researchers.
**Reviewer #2 (Public review):**
Summary:This study elucidated the mechanism underlying drug resistance induced by CDK4/6i as a single agent and proposed a novel and efficacious second-line therapeutic strategy. It highlighted the potential of combining CDK2i with CDK4/6i for the treatment of HR+/HER2- breast cancer.Strengths:The study demonstrated that CDK4/6 induces drug resistance by impairing Rb activation, which results in diminished E2F activity and a delay in G1 phase progression. It suggests that the synergistic use of CDK2i and CDK4/6i may represent a promising second-line treatment approach. Addressing critical clinical challenges, this study holds substantial practical implications.Weaknesses:(1) Drug-resistant cell lines: Was a drug concentration gradient treatment employed to establish drug-resistant cell lines? If affirmative, this methodology should be detailed in the materials and methods section.

We greatly appreciate the reviewer for raising this important question. In the revised manuscript, we have updated the methods section (“Drug-resistant cell lines”) to more precisely describe how the drug-resistant cell lines were established.

(2) What rationale informed the selection of MCF-7 cells for the generation of CDK6 knockout cell lines? Supplementary Figure 3. A indicates that CDK6 expression levels in MCF-7 cells are not notably elevated.

We appreciate the reviewer’s insightful question about the rationale for selecting MCF-7 cells to generate CDK6 knockout cell lines. This choice was guided by prior studies highlighting the significant role of CDK6 in mediating resistance to CDK4/6 inhibitors (21-24). Moreover, we observed a 4.6-fold increase in CDK6 expression in CDK4/6i resistant MCF-7 cells compared to their drug-naïve counterparts (Supplementary Figure 3A). While we did not detect notable differences in CDK4/6 activity between wild-type and CDK6 knockout cells under CDK4/6 inhibitor treatment, these findings point to a potential non-canonical function of CDK6 in conferring resistance to CDK4/6 inhibitors.

(3) For each experiment, particularly those involving mice, the author must specify the number of individuals utilized and the number of replicates conducted, as detailed in the materials and methods section.

We sincerely thank the reviewer for bringing this to our attention. In the revised manuscript, we have explicitly stated the number of replicates and mice used for each experiment as appropriate in figure legends and relevant text to ensure transparency and clarity.

(4) Could this treatment approach be extended to triple-negative breast cancer?

We greatly appreciate the reviewer’s inquiry about extending our findings to triple-negative breast cancer (TNBC). Based on the data presented in Figure 1 and Supplementary Figure 2, which include the TNBC cell line MDA-MB-231, we expect that the benefits of maintaining CDK4/6 inhibitors could indeed be applicable to TNBC with an intact Rb/E2F pathway. Additionally, our recent paper (25) indicates a similar mechanism in TNBC.

**Reviewer #3 (Public review):**
Summary:In their manuscript, Armand and colleagues investigate the potential of continuing CDK4/6 inhibitors or combining them with CDK2 inhibitors in the treatment of breast cancer that has developed resistance to initial therapy. Utilizing cellular and animal models, the research examines whether maintaining CDK4/6 inhibition or adding CDK2 inhibitors can effectively control tumor growth after resistance has set in. The key findings from the study indicate that the sustained use of CDK4/6 inhibitors can slow down the proliferation of cancer cells that have become resistant, and the combination of CDK2 inhibitors with CDK4/6 inhibitors can further enhance the suppression of tumor growth. Additionally, the study identifies that high levels of Cyclin E play a significant role in resistance to the combined therapy. These results suggest that continuing CDK4/6 inhibitors along with the strategic use of CDK2 inhibitors could be an effective strategy to overcome treatment resistance in hormone receptor-positive breast cancer.Strengths:(1) Continuous CDK4/6 Inhibitor Treatment Significantly Suppresses the Growth of Drug-Resistant HR+ Breast Cancer: The study demonstrates that the continued use of CDK4/6 inhibitors, even after disease progression, can significantly inhibit the growth of drug-resistant breast cancer.(2) Potential of Combined Use of CDK2 Inhibitors with CDK4/6 Inhibitors: The research highlights the potential of combining CDK2 inhibitors with CDK4/6 inhibitors to effectively suppress CDK2 activity and overcome drug resistance.(3) Discovery of Cyclin E Overexpression as a Key Driver: The study identifies overexpression of cyclin E as a key driver of resistance to the combination of CDK4/6 and CDK2 inhibitors, providing insights for future cancer treatments.(4) Consistency of In Vitro and In Vivo Experimental Results: The study obtained supportive results from both in vitro cell experiments and in vivo tumor models, enhancing the reliability of the research.(5) Validation with Multiple Cell Lines: The research utilized multiple HR+/HER2- breast cancer cell lines (such as MCF-7, T47D, CAMA-1) and triple-negative breast cancer cell lines (such as MDA-MB-231), validating the broad applicability of the results.Weaknesses:(1) The manuscript presents intriguing findings on the sustained use of CDK4/6 inhibitors and the potential incorporation of CDK2 inhibitors in breast cancer treatment. However, I would appreciate a more detailed discussion of how these findings could be translated into clinical practice, particularly regarding the management of patients with drug-resistant breast cancer.

Thank you to the reviewer for this crucial comment. In the revised Discussion, we've broadened our exploration of clinical translation. Specifically, we emphasize that ongoing CDK4/6 inhibition, although not fully stopping resistant tumors, significantly slows their growth and may offer a therapeutic window when combined with ET and CDK2 inhibition. We also note that these approaches may work best for patients without Rb loss or newly acquired resistance-driving mutations, and that cyclin E overexpression could be a biomarker to inform patient selection. These points together highlight that our findings provide a mechanistic understanding and potential framework for clinical trials testing maintenance CDK4/6i with selective addition of CDK2i as a secondline strategy in drug-resistant HR+/HER2- breast cancer.

(2) While the emergence of resistance is acknowledged, the manuscript could benefit from a deeper exploration of the molecular mechanisms underlying resistance development. A more thorough understanding of how CDK2 inhibitors may overcome this resistance would be valuable.

We thank the reviewer for this valuable suggestion. In the revised manuscript, we have expanded our Discussion to more explicitly synthesize the molecular mechanisms of resistance and how CDK2 inhibitors counteract them. Specifically, we describe how sustained CDK4/6 inhibition drives a non-canonical route of Rb degradation, resulting in inefficient E2F activation and prolonged G1 phase progression. We also highlight the role of c-Myc in amplifying E2F activity and promoting resistance, and we show that continued ET mitigates this effect by suppressing c-Myc. Importantly, we demonstrate that CDK2 inhibition alone cannot fully suppress the growth of resistant cells, but when combined with CDK4/6 inhibition, it produces durable repression of E2F and Myc target gene programs and significantly delays the G1/S transition. Finally, we identify cyclin E overexpression as a key mechanism of escape from dual CDK4/6i + CDK2i therapy, suggesting its potential as a biomarker for patient stratification . Together, these findings provide a detailed mechanistic rationale for how CDK2 inhibition can overcome specific pathways of resistance in HR^+^/HER2^-^ breast cancer.

(3) The manuscript supports the continued use of CDK4/6 inhibitors, but it lacks a discussion on the long-term efficacy and safety of this approach. Additional studies or data to support the safety profile of prolonged CDK4/6 inhibitor use would strengthen the manuscript.

We appreciate the reviewer’s insightful comment. In the revised manuscript, we emphasize the longterm efficacy and safety considerations of sustained CDK4/6 inhibition. Clinical trial and retrospective data have shown that continued CDK4/6i therapy can extend progression-free survival in selected patients, while maintaining a favorable safety profile (26-28). We have updated the Discussion to highlight these findings more explicitly, underscoring that while prolonged CDK4/6 inhibition slows but does not fully arrest tumor growth, it remains a clinically viable strategy when balanced against its manageable toxicity profile.

**Reviewer #1 (Recommendations for the authors):**
It is well known that the combination therapy of CDK4/6i and ET has therapeutic benefits in ER(+) HER2(-) advanced breast cancer. However, drug resistance is a problem, and second-line therapy to solve this problem has not been established. Although some parts of the research results are already reported, the authors confirmed them by employing live cell markers, and further proved and suggested how to overcome this resistance in detail. This part is considered novel.Overall, this research manuscript is eligible to be accepted with the appropriate addressing of questions.(1)The effects and biochemical changes of combination therapy of CDK4/6i and CDK2i are already known in several papers. The author needs to highlight the differences between the author's research and that of otherresearchers.

We thank the reviewer for the opportunity to clarify the novelty of our findings in the context of prior studies on CDK4/6i and CDK2i combination therapy. In the revised manuscript, we have updated the Discussion section to more clearly delineate how our work extends and differs from existing research.

Specifically, we now state:

Page 12: The combination of CDK4/6i and ET has reshaped treatment for HR^+^/HER2^-^ breast cancer (1-8). However, resistance commonly emerges, and no consensus second-line standard is established. Our data show that continued CDK4/6i treatment in drug-resistant cells engages a non-canonical, proteolysis-driven route of Rb inactivation, yielding attenuated E2F output and a pronounced delay in G1 progression (Figure 7G). Concurrent ET further deepens this blockade by suppressing c-Myc-mediated E2F amplification, thereby prolonging G1 and slowing population growth. Importantly, CDK2 inhibition alone was insufficient to control resistant cells. Robust suppression of CDK2 activity and resistant-cell growth required CDK2i in combination with CDK4/6i, consistent with prior reports supporting dual CDK targeting (9-16). Moreover, cyclin E, and in some contexts cyclin A, blunted the efficacy of the CDK4/6i and CDK2i combination by reactivating CDK2. Together, these findings provide a mechanistic rationale for maintaining CDK4/6i beyond progression and support testing ET plus CDK4/6i with the strategic addition of CDK2i, as evidenced by concordant in vitro and in vivo results.

(2) Regarding Figures 3H and 3I, I wonder if it is live cell imaging results or if the authors counter each signal via timed IF staining slides? If live cell imaging is used, the authors need to present the methods.

We appreciate the reviewer’s question. Figures 3H and 3I derive from a live–fixed correlative pipeline rather than purely live imaging or independently timed IF slides. We first imaged asynchronously proliferating cells live for ≥48 h to (i) segment/track nuclei with H2B fluorescence, (ii) define mitotic exit (t = 0 at anaphase), and (iii) record CDK2 activity using a CDK2 KTR in the last live frame. Immediately after the live acquisition, we pulsed EdU (10 µM, 15 min) and fixed the same wells, photobleached fluorescent proteins (3% H₂O₂ + 20 mM HCl, 2 h, RT) to prevent crosstalk, and then performed click-chemistry EdU detection, IF for phospho-Rb (Ser807/811) and total Rb, and RNA FISH for E2F1. Fixed-cell readouts (p-Rb positivity, EdU incorporation, E2F1 mRNA puncta) were mapped back to each single cell’s live-derived time since mitosis and/or CDK2 activity, enabling the kinetic plots shown in Fig. 3H–I.

To ensure transparency and reproducibility, we added detailed methods describing this workflow in the “Immunofluorescence and mRNA fluorescence in situ hybridization (FISH)” section under a dedicated “live– fixed pipeline” paragraph, and we cross-referenced acquisition and analysis parameters in “Live- and fixed-cell image acquisition” and “Image processing and analysis.” These updates specify: EdU pulse/fix conditions, photobleaching, antibodies/probes, imaging hardware and channels, segmentation/tracking, mitosis alignment, background correction, and how fixed readouts were binned/quantified as functions of time after mitosis and CDK2 activity.

(3) Regarding Figure 3F, seven images were obtained in same fields? The author needs to describe the meaning of the white image and the yellow and blue image of the bottom in detail.

Thank you for raising this point. All seven panels in Fig. 3F are from the same field of view. The top row shows the raw channels (Hoechst, p-Rb, total Rb, and E2F1 RNA FISH). The bottom row shows the corresponding processed outputs from that field: (i) nuclear segmentation, (ii) phosphorylated Rb-status classification, and (iii) cell boundaries used for single-cell RNA-FISH quantification. We have revised the figure legend to make this explicit.

(4) The author showed E2F mRNA by ISH, but in fact, RB does not suppress E2F mRNA but suppresses protein, so the author needs to confirm E2F at the protein level.

We sincerely appreciate the reviewer’s thoughtful suggestion to examine E2F1 at the protein level. In our study, we focused on E2F1 mRNA expression because it is a well-established and biologically meaningful readout of E2F1 transcriptional activity. Due to its autoregulatory nature (17), the release of active E2F1 protein from Rb induces the transcription of E2F1 itself, creating a positive feedback loop. As a result, E2F1 mRNA abundance serves as a direct and reliable proxy for E2F1 protein activity (18-20). Thus, quantifying E2F1 mRNA provides a biologically relevant and mechanistic indicator of Rb-E2F pathway status. To clarify this rationale, we have updated the Results section and added references supporting our use of E2F1 mRNA as a readout for E2F1 activity.

(5) Is it possible to synchronize cells (nocodazole shake-off, Double thymidine block) under the presence of cdk4/6i? If so, then the authors need to demonstrate the delay of G1 progression via immunoblotting.

We thank the reviewer for this constructive suggestion. To address it, we performed nocodazole synchronization followed by release and monitored cell-cycle progression in the presence or absence of CDK4/6 inhibition.

Specifically, we added the following new datasets to the revised manuscript:

Fig. 3L: Live single-cell trajectories of CDK4/6 and CDK2 activities alongside the Cdt1-degron reporter after 14 hours of nocodazole (250 nM) treatment and release. We compared the averaged traces of CDK4/6 and CDK2 activities and Cdt1 intensity in parental cells (gray) and resistant cells with (red) and without (blue) CDK4/6i maintenance. These data show suppressed and delayed CDK2 activation, as well as a right-shifted S-phase entry, particularly under continuous CDK4/6 inhibition.

Fig. 3M: Fixed-cell EdU pulse-labeling at 4, 6, 8, 12, 16, and 24 h post-release further confirms a significant delay in S-phase entry and prolonged G1 duration in CDK4/6i-maintained cells compared with naïve and withdrawn conditions.

Together, these results directly demonstrate the delay in G1 progression following synchronized mitotic exit under CDK4/6 inhibition.

(6) In Figure 5C the authors showed a violin plot of c-Myc level. Is this Immunohistochemical staining? The authors need to clarify the methods.

Thank you for flagging this. The c-Myc measurements in Fig. 5C are from immunofluorescence (IF), not IHC. We now state this explicitly in the legend.

(7) Regarding Live cell immunofluorescence tracing of live-cell reporters, the author needs to clarify the methods (excitation, emission), name of instruments, and software used.

To address this, we have expanded the “Live-cell, fixed-cell, and tumor tissue image acquisition” section in the Materials and Methods.

(8) Lines 475 SF1A, the authors need to correct typos. Naïve Naïve.

We greatly appreciate the reviewer’s attention to this detail and have ensured all typos have been addressed.

(9) The authors need to unify Cdt1-degron(legends) Vs Cdt1 degron (figures).

We greatly appreciate your attention to this discrepancy. Language referring to the Cdt1 degron has been unified between figures and legends.

**Reviewer #3 (Recommendations for the authors):**
(1) While the manuscript discusses the selection of doses for CDK4/6 inhibitors and CDK2 inhibitors, there is a lack of detailed data on the dose-response relationship. Additional data on the effects of different doses would be beneficial.

We appreciate the reviewer’s important comment. To address it, we performed additional dose– response experiments testing a range of CDK4/6i and CDK2i concentrations. These analyses revealed a clear synergistic interaction between the two inhibitors. The new data are now presented in Figure 6G and Supplementary Figure 8F of the revised manuscript.

(2) In clinical trials, the criteria for patient selection are crucial for interpreting study outcomes. A detailed description of the patient selection criteria should be provided.

We thank the reviewer for bringing this important point to our attention. In the revised manuscript, we have clarified the patient selection criteria relevant to the interpretation of clinical outcomes. Specifically, we note that retrospective analyses suggest patients with indolent disease and no prior chemotherapy may benefit most from continued CDK4/6i plus ET. Moreover, our data and others’ indicate that clinical benefit is expected in tumors retaining an intact Rb/E2F axis, while resistance-driving alterations (e.g., Rb loss, PIK3CA, ESR1, FGFR1–3, HER2, FAT1 mutations) are likely to limit efficacy. Finally, we highlight cyclin E overexpression as a potential biomarker of resistance to combined CDK4/6i and CDK2i, underscoring the need for biomarker-guided patient stratification. These additions provide a more detailed framework for patient selection in future clinical applications.

References

(1) Finn RS, Crown JP, Lang I, Boer K, Bondarenko IM, Kulyk SO, et al. The cyclin-dependent kinase 4/6 inhibitor palbociclib in combination with letrozole versus letrozole alone as first-line treatment of oestrogen receptor-positive, HER2-negative, advanced breast cancer (PALOMA-1/TRIO-18): a randomised phase 2 study. Lancet Oncol 2015;16:25-35

(2) Finn RS, Martin M, Rugo HS, Jones S, Im S-A, Gelmon K, et al. Palbociclib and Letrozole in Advanced Breast Cancer. New England Journal of Medicine 2016;375:1925-36

(3) Turner NC, Slamon DJ, Ro J, Bondarenko I, Im S-A, Masuda N, et al. Overall Survival with Palbociclib and Fulvestrant in Advanced Breast Cancer. New England Journal of Medicine 2018;379:1926-36

(4) Dickler MN, Tolaney SM, Rugo HS, Cortés J, Diéras V, Patt D, et al. MONARCH 1, A Phase II Study of Abemaciclib, a CDK4 and CDK6 Inhibitor, as a Single Agent, in Patients with Refractory HR(+)/HER2(-) Metastatic Breast Cancer. Clin Cancer Res 2017;23:5218-24

(5) Johnston S, Martin M, Di Leo A, Im S-A, Awada A, Forrester T, et al. MONARCH 3 final PFS: a randomized study of abemaciclib as initial therapy for advanced breast cancer. npj Breast Cancer 2019;5:5

(6) Hortobagyi GN, Stemmer SM, Burris HA, Yap Y-S, Sonke GS, Hart L, et al. Overall Survival with Ribociclib plus Letrozole in Advanced Breast Cancer. New England Journal of Medicine 2022;386:94250

(7) Slamon DJ, Neven P, Chia S, Fasching PA, De Laurentiis M, Im S-A, et al. Overall Survival with Ribociclib plus Fulvestrant in Advanced Breast Cancer. New England Journal of Medicine 2019;382:51424

(8) Im S-A, Lu Y-S, Bardia A, Harbeck N, Colleoni M, Franke F, et al. Overall Survival with Ribociclib plus Endocrine Therapy in Breast Cancer. New England Journal of Medicine 2019;381:307-16

(9) Pandey K, Park N, Park KS, Hur J, Cho YB, Kang M, et al. Combined CDK2 and CDK4/6 Inhibition Overcomes Palbociclib Resistance in Breast Cancer by Enhancing Senescence. Cancers (Basel) 2020;12

(10) Freeman-Cook K, Hoffman RL, Miller N, Almaden J, Chionis J, Zhang Q, et al. Expanding control of the tumor cell cycle with a CDK2/4/6 inhibitor. Cancer Cell 2021;39:1404-21 e11

(11) Dietrich C, Trub A, Ahn A, Taylor M, Ambani K, Chan KT, et al. INX-315, a selective CDK2 inhibitor, induces cell cycle arrest and senescence in solid tumors. Cancer Discov 2023

(12) Al-Qasem AJ, Alves CL, Ehmsen S, Tuttolomondo M, Terp MG, Johansen LE, et al. Co-targeting CDK2 and CDK4/6 overcomes resistance to aromatase and CDK4/6 inhibitors in ER+ breast cancer. NPJ Precis Oncol 2022;6:68

(13) Kudo R, Safonov A, Jones C, Moiso E, Dry JR, Shao H, et al. Long-term breast cancer response to CDK4/6 inhibition defined by TP53-mediated geroconversion. Cancer Cell 2024

(14) Arora M, Moser J, Hoffman TE, Watts LP, Min M, Musteanu M, et al. Rapid adaptation to CDK2 inhibition exposes intrinsic cell-cycle plasticity. Cell 2023;186:2628-43 e21

(15) Kumarasamy V, Wang J, Roti M, Wan Y, Dommer AP, Rosenheck H, et al. Discrete vulnerability to pharmacological CDK2 inhibition is governed by heterogeneity of the cancer cell cycle. Nature Communications 2025;16:1476

(16) Dommer AP, Kumarasamy V, Wang J, O'Connor TN, Roti M, Mahan S, et al. Tumor Suppressors Condition Differential Responses to the Selective CDK2 Inhibitor BLU-222. Cancer Res 2025

(17) Johnson DG, Ohtani K, Nevins JR. Autoregulatory control of E2F1 expression in response to positive and negative regulators of cell cycle progression. Genes & Development 1994;8:1514-25

(18) Chung M, Liu C, Yang HW, Koberlin MS, Cappell SD, Meyer T. Transient Hysteresis in CDK4/6 Activity Underlies Passage of the Restriction Point in G1. Mol Cell 2019;76:562-73 e4

(19) Kim S, Leong A, Kim M, Yang HW. CDK4/6 initiates Rb inactivation and CDK2 activity coordinates cell-cycle commitment and G1/S transition. Sci Rep 2022;12:16810

(20) Yang HW, Chung M, Kudo T, Meyer T, Yang HW, Chung, Mingyu, Kudo T, et al. Competing memories of mitogen and p53 signalling control cell-cycle entry. Nature 2017;549:404-8

(21) Yang C, Li Z, Bhatt T, Dickler M, Giri D, Scaltriti M, et al. Acquired CDK6 amplification promotes breast cancer resistance to CDK4/6 inhibitors and loss of ER signaling and dependence. Oncogene 2017;36:2255-64

(22) Li Q, Jiang B, Guo J, Shao H, Del Priore IS, Chang Q, et al. INK4 Tumor Suppressor Proteins Mediate Resistance to CDK4/6 Kinase Inhibitors. Cancer Discov 2022;12:356-71

(23) Ji W, Zhang W, Wang X, Shi Y, Yang F, Xie H, et al. c-myc regulates the sensitivity of breast cancer cells to palbociclib via c-myc/miR-29b-3p/CDK6 axis. Cell Death & Disease 2020;11:760

(24) Wu X, Yang X, Xiong Y, Li R, Ito T, Ahmed TA, et al. Distinct CDK6 complexes determine tumor cell response to CDK4/6 inhibitors and degraders. Nature Cancer 2021;2:429-43

(25) Kim S, Son E, Park HR, Kim M, Yang HW. Dual targeting CDK4/6 and CDK7 augments tumor response and anti-tumor immunity in breast cancer models. J Clin Invest 2025

(26) Ravani LV, Calomeni P, Vilbert M, Madeira T, Wang M, Deng D, et al. Efficacy of Subsequent Treatments After Disease Progression on CDK4/6 Inhibitors in Patients With Hormone Receptor-Positive Advanced Breast Cancer. JCO Oncol Pract 2025;21:832-42

(27) Martin JM, Handorf EA, Montero AJ, Goldstein LJ. Systemic Therapies Following Progression on Firstline CDK4/6-inhibitor Treatment: Analysis of Real-world Data. Oncologist 2022;27:441-6

(28) Kalinsky K, Bianchini G, Hamilton E, Graff SL, Park KH, Jeselsohn R, et al. Abemaciclib Plus Fulvestrant in Advanced Breast Cancer After Progression on CDK4/6 Inhibition: Results From the Phase III postMONARCH Trial. J Clin Oncol 2025;43:1101-12